# Detection of water vapour absorption around 363 nm in measured atmospheric absorption spectra and its effect on DOAS evaluations

Johannes Lampel[1,*], Denis Pöhler[2], Oleg L. Polyansky[3,4], Aleksandra A. Kyuberis[4], Nikolai F. Zobov[4], Jonathan Tennyson[3], Lorenzo Lodi[3], Udo Frieß[2], Yang Wang[1], Steffen Beirle[1], Ulrich Platt[2], and Thomas Wagner[1]

[1]Max Planck Institute for Chemistry, 55128 Mainz, Germany
[2]Institute of Environmental Physics, University of Heidelberg, 69120 Heidelberg, Germany
[3]Department of Physics and Astronomy, University College London, Gower St, London WC1E 6BT, UK
[4]Institute of Applied Physics, Russian Academy of Sciences, Nizhny Novgorod, Russia
[*]now at [2]

*Correspondence to:* J. Lampel (johannes.lampel@mpic.de)

**Abstract.**

Water vapour is known to absorb radiation from the microwave region to the blue part of the visible spectrum at a decreasing efficiency. Ab-initio approaches to model individual absorption lines of the gaseous water molecule predict absorption lines until its dissociation limit at 243 nm.

We present first evidence of water vapour absorption near 363 nm from field measurements using data from Multi-Axis differential optical absorption spectroscopy (MAX-DOAS) and Longpath (LP)-DOAS measurements. The identification of the absorptions was based on the recent *POKAZATEL* line list by Polyansky et al. (2016).

We observed absorption by water vapour at 363 nm with optical depths of up to $2 \times 10^{-3}$. It correlates well with simultaneously measured well-established water vapour absorptions in the blue spectral range from 452-499 nm ($R^2 = 0.89$), but the line intensities are underestimated by a factor of $2.6 \pm 0.5$ by the ab-initio model. At a spectral resolution of 0.5 nm, we derive a maximum cross-section value of $2.7 \times 10^{-27}$ cm$^2$ molec$^{-1}$ at 362.3 nm.

The newly found absorption can have a significant impact on the spectral retrieval of absorbing trace-gas species in the spectral range around 363 nm. Its effect on the spectral analysis of $O_4$, HONO and OClO is discussed.

## 1 Introduction

The most important greenhouse gas is water vapour. it plays a key role for the radiative balance of the Earth's atmosphere (e.g. Myhre et al., 2013). Due to the large temperature range covered by observations on Earth, the upper atmosphere but also on exoplanets, and due to the spectral extend of observed water vapour absorption, accurate water vapour line lists covering different temperatures over a wide range of wavelengths are necessary. Since water vapour absorptions are present in many wavelength regions, precise knowledge of their properties is also required assessing greenhouse effects. In addition it is required for spectroscopic detection of other trace gases, since their absorption structures often overlap with water vapour absorption. The number of laboratory measurements of water vapour absorption spectra at different temperatures is limited due to technical

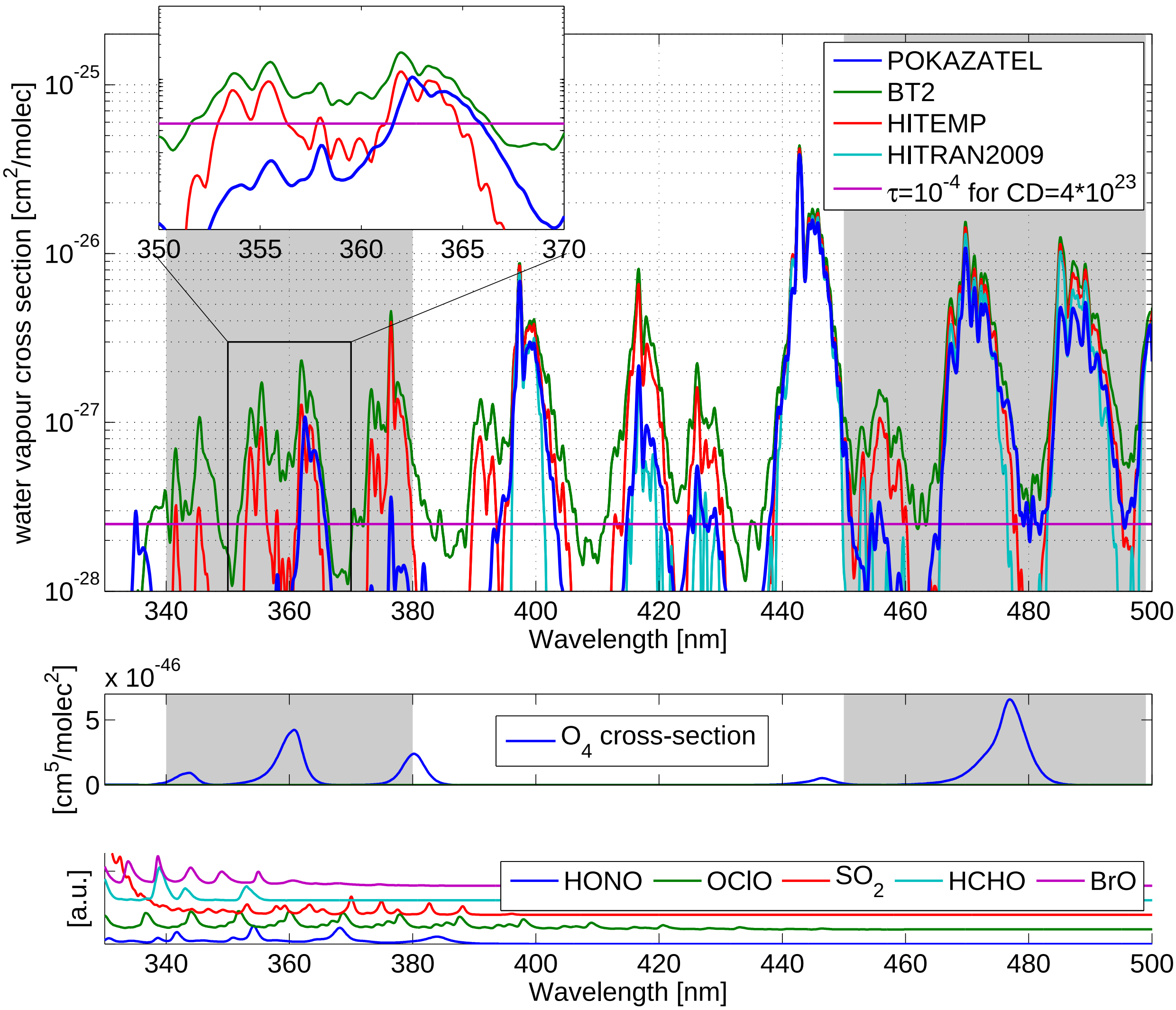

**Figure 1.** Overview of some recently published published water vapour cross-sections convoluted to a spectral resolution of 0.5 nm in the spectral interval from 330 to 500 nm. Also indicated is a typical MAX-DOAS detection limit for a differential OD of $10^{-4}$ at a water vapour column density of $4 \cdot 10^{23}$ molec cm$^{-2}$ (purple line, top panel). The middle panels shows the $O_4$ absorption cross-section, the lowermost panel other absorbers of atmospheric relevance (HONO, OClO, $SO_2$, HCHO, and BrO) in this spectral range.

reasons: Experimental measurements of water vapour absorption are not straightforward, as water vapour cannot be compressed to increase its optical depths in a measurement volume at any temperature. Moreover the absorption cross-section is relatively small in certain wavelength ranges, e.g. in the blue and near UV spectral ranges which concern us here. The gap between observed absorptions and the available literature absorption cross-sections from laboratory measurements can be addressed by means of *ab initio* models for water vapour absorption lines, which can provide energy (i.e. wavelength), intensity, and additional parameters for each absorption line. This is done e.g. in the HITRAN database (Rothman et al., 2013), where information from measured absorption lines is merged with information from other sources such as *ab initio* models. In addition to HITRAN, other line list compilations are also available such as the GEISA database (Jacquinet-Husson et al., 2008), which lists water vapour absorption lines up to 25232 $cm^{-1}$ (down to 396.3 nm).

Polyansky et al. (2016) recently developed a computed line list (which we call *POKAZATEL* here, according to the first letters of the name of each author) containing water vapour lines in the spectral range below 400 nm. Prior to this publication, absorptions in this spectral region were already listed in the computed line list BT2 (Barber et al., 2006), on which HITEMP (Rothman et al., 2010) is partly based. However, only a few of these absorption lines have also been reported from laboratory measurements (Dupré et al., 2005; Maksyutenko et al., 2012). For a compilation of spectroscopic data see Tennyson et al. (2013). HITRAN 2012 (Rothman et al., 2013) does not list absorptions below 388 nm.

Recently, Wilson et al. (2016) deduced upper limits for the water vapour absorption in the near-UV by incoherent broadband cavity enhanced absorption spectroscopy (IBBCEAS) measurements. They estimated the water vapour absorption cross-section to be smaller than $5 \times 10^{-26} cm^2$ $molec^{-1}$ at a spectral resolution of 0.5 nm between 340–420 nm. This is significantly smaller than the water vapour cross-section measured by Du et al. (2013) between 290–350 nm (see subsection 4.8).

Lampel et al. (2015b) found systematic residual structures in this spectral range below 370 nm with magnitudes of around $5 \times 10^{-4}$ in Multi-Axis Differential Optical Absorption Spectroscopy (MAX-DOAS) atmospheric observations, which could point towards a tropospheric absorber with absorption structures in this spectral range. The BT2 and HITEMP line lists could explain some of the structures, but show inconsistencies. These two line lists show significant differences between each other, mostly due to the individual line cut-off employed in the HITEMP database (see also Figure 1).

### 1.1 The '*POKAZATEL*' line list

Following up on previous high quality computed water line lists (Partridge and Schwenke, 1997; Barber et al., 2006), the *POKAZATEL* line list (Polyansky et al., 2016) was calculated for the purpose of producing a complete list of water lines involving transitions between all the bound energy levels of $H_2{}^{16}O$ up to dissociation.

Until now the most complete water line list, called BT2 (Barber et al., 2006), only covered energy levels up to 30 000 $cm^{-1}$ (333 nm) and rotational quantum numbers, $J$, up to 50. POKAZATEL covers the entire bound energies up to dissociation – 41 000 $cm^{-1}$ (244 nm) (Boyarkin et al., 2013) and the highest $J$ considered is 72.

*POKAZATEL* extends BT2 threefold. First, higher temperatures can be covered by the line list, as higher energy levels are involved and more hot transitions are calculated. Second, for room temperature the spectral range is expanded in the UV region down to (in principle) about 244 nm. Third, the predictions of the line positions and intensities by *POKAZATEL* should be

considerably more accurate. In particular, POKAZATEL is based on variational nuclear motion calculations performed with the DVR3D program suite (Tennyson et al., 2004). In order to calculate the line positions and line intensities of the water lines two inputs into DVR3D are necessary – a water potential energy surface (PES) for the ground electronic state and a dipole moment surface (DMS). A global water PES, covering geometries up to dissociation, is available only from *ab initio* calculations (Császár et al., 2010) and is not accurate enough for our purposes. *POKAZATEL* is therefore based on the semi-empirical PES obtained by the fitting to the experimental data up to 41 000 $cm^{-1}$ (Tennyson et al., 2013). The details of the fit are given by Polyansky et al. (2016). In particular, the RMS (root mean square) deviation for levels below 25 000 $cm^{-1}$, calculated by this fitted PES is about 0.03 $cm^{-1}$ and the levels from 25 000 $cm^{-1}$ to 41 000 $cm^{-1}$ are reproduced to within about 0.1 $cm^{-1}$ on average.

A very accurate, *ab initio*, global DMS was computed by Lodi et al. (2011) and was used without modification for the POKAZATEL line list calculation. This DMS has been used to successfully construct comprehensive line lists for $H_2{}^{17}O$ and $H_2{}^{18}O$ (Lodi and Tennyson, 2012) which were included in their entirity in the most recent, 2012, release of HITRAN. A recent laboratory investigation has verified the accuracy of these line lists in the near-infrared (Regalia et al., 2014). However, as discussed below, the intensities predicted by the various line lists have yet to be validated in the near-uv.

### 1.2 Potential impact on DOAS measurements of atmospheric trace gases

The absorption lines listed in the UV range in *POKAZATEL*, BT2 and HITEMP - which are to our knowledge presently not included in DOAS retrievals - could have an effect on the overall measurement errors of several trace-gas retrievals and could lead to systematic biases in the spectral evaluation of tropospheric absorbers in this spectral regions, such as $O_4$, HONO, OClO and $SO_2$, potentially even HCHO and BrO. In subsection 4.11 we discuss these potential interferences.

In particular, spectral structures at around 360 nm have been observed in atmospheric DOAS measurements before and were either explained by erroneous oxygen dimer $O_2$-$O_2$ (or in short: $O_4$) literature cross-sections, e.g. an incorrect spectral calibration of the respectively used cross-section data (e.g. Wagner et al., 2002), or by an unaccounted tropospheric absorber. *POKAZATEL* lists a number of absorption lines which overlay the $O_4$ absorption at 360 nm. In cases where the absolute humidity during the measurement campaign does not change and therefore the correlation of the column densities determined from the absorptions of water vapour and $O_4$ is strong, as it is the case e.g. for the M91 campaign (see section 2), it is difficult to disentangle the possible contribution of water vapour absorption and $O_4$ absorption.

### 1.3 Outline

Based on our field measurements combined with the *POKAZATEL* water vapour line list, which yields new information about water vapour absorption below 390nm, we make an attempt to answer the following questions:

1. Are the water vapour absorption bands near 335 nm, 363 nm and 376 nm found in atmospheric DOAS measurements?
2. Is the magnitude of these absorptions in agreement with measurements in other wavelength ranges? (compare also Lampel et al., 2015b, for the blue spectral range)

3. How well is the shape and the magnitude of the measured absorption bands reproduced by the line lists?
4. What are the consequences for the spectral retrieval of other trace-gases in the same spectral region? (as e.g. $O_4$, HONO and OClO)

## 2 Atmospheric DOAS measurements

The data which was analysed here was collected during three different field campaigns, where different DOAS instruments were used.

1. MAX-DOAS data from cruises *ANT XXVIII/1-2* (Naggar, 2012; Kattner, 2012) of the research vessel 'Polarstern', which covered latitudes from 54°N (northern Germany) to 70°S (coastal Antarctica).
2. MAX-DOAS data from the 'Surface Ocean PRocesses in the ANtropocene' (SOPRAN) cruise M91 with the research vessel 'Meteor' in the Peruvian upwelling region in December 2012 (Bange, 2013).
3. Longpath (LP)-DOAS Measurements were analysed for water vapour using data from a dedicated measurement period in Heidelberg in August and September 2015 (further called *HD15* ).

Both MAX-DOAS cruises were largely unaffected by anthropogenic pollution, which avoids interferences of high $NO_2$ absorption structures in the data evaluation.

The MAX-DOAS measurements during the M91 campaign were performed at a spectral resolution of 0.45 nm, but due to the limited latitudinal extent of the cruise track (compare Table 1 and Figure 2), the variation of water vapour volume mixing ratios (VMR) was small. The VMR was, according to the meteorological station onboard, between 1.6 – 2.4%. Therefore observed differential column densities (dSCDs, section 3) of $H_2O$ and $O_4$ correlate well due to changes in the effective light path lengths and cannot be unambiguously disentangled. The campaign *ANT XXVIII/1-2* took place along a cruise track from Bremerhaven/Germany to Antarctica and therefore allows to distinguish actual water vapour absorption from systematic errors in the $O_4$ cross-sections employed. Water vapour VMR were found between 0.5 – 3.0% according to the meteorological station onboard. The MAX-DOAS instrument onboard Polarstern has a lower spectral resolution of 0.7 nm (UV) and 0.9 nm (VIS), but has the advantage of a wider spectral range allowing for independent simultaneous observations of $H_2O$ and $O_4$ at around 361 nm and 477 nm due to the spectral overlap of both absorbers.

Dedicated LP-DOAS measurements were performed in Heidelberg in August and September 2015. The advantage is the high spectral resolution of 0.2 nm and the well-defined light-path. However, high $NO_2$ concentrations can cause spectral interferences and the range of absolute water vapour volume mixing ratios (VMR) is relatively limited (see subsection 4.1).

## 3 The DOAS Method

The DOAS (Differential Optical Absorption Spectroscopy) method (Platt and Stutz, 2008) relies on attenuation of light from suitable light sources (intensity $I_0$) by absorbers within the light path according to Lambert-Beer's law $I(\lambda) =$

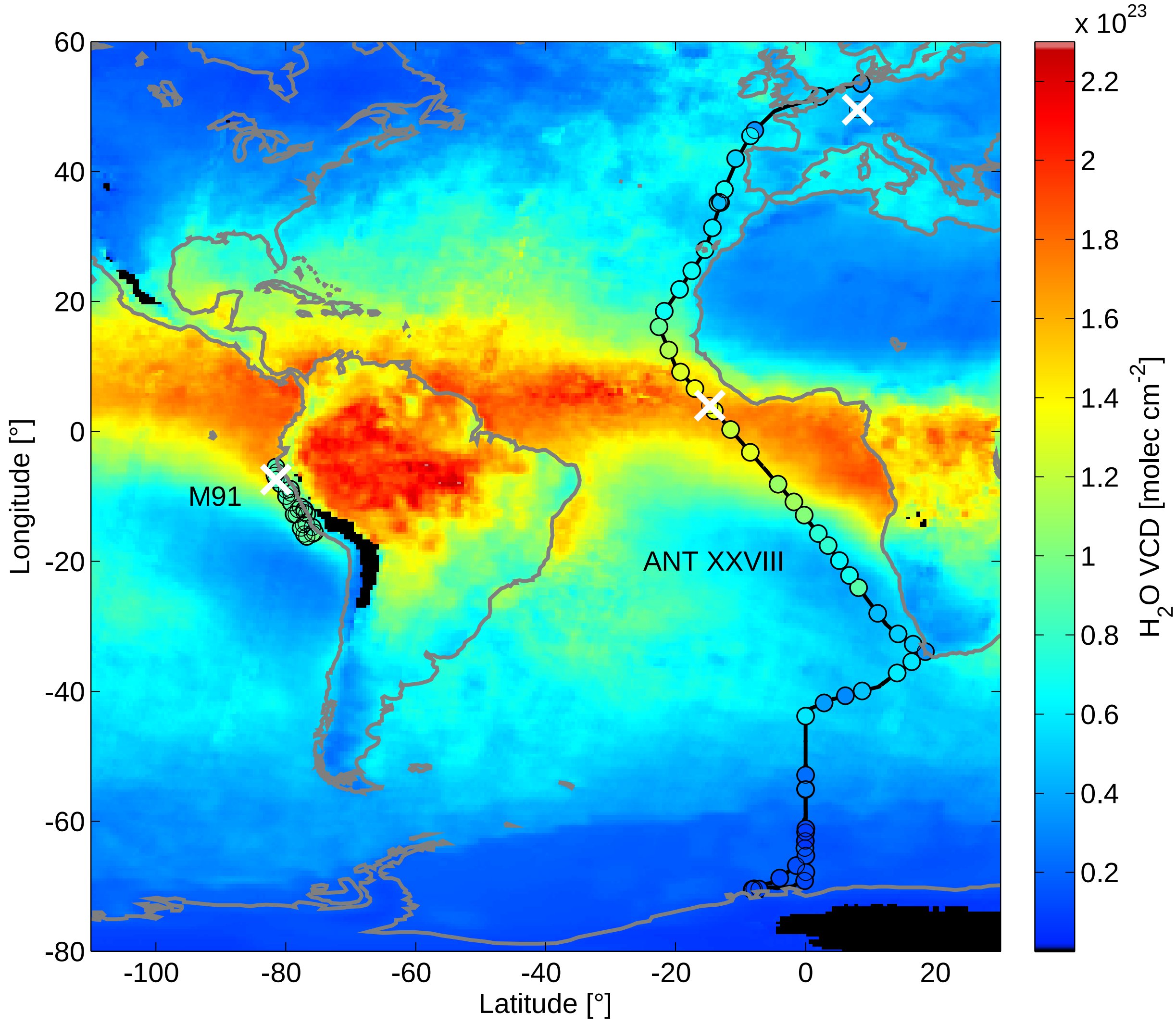

**Figure 2.** Measurement Overview: The cruise track of M91 (Peruvian Upwelling) and *ANT XXVIII/1-2* (Atlantic) is shown, additionally the location of the LP-DOAS measurements in Heidelberg, Germany is marked (white cross in the north-east corner of the map). The background shows GOME-2A $H_2O$ VCDs (Wagner et al., 2003) averaged from November and December 2011 (time during *ANT XXVIII/1-2* 12). The locations of the measurements shown in Figure 3 and Figure 5 are also marked by white crosses. Daily error-weighted averages of $H_2O/O_4$ dSCD ratios (measured in the wavelength range from 340-380 nm, corrected according to Figure 6) are shown as circles and converted to a VCD assuming an exponential water vapour concentration profile with a scale height of 2 km.

| Name | Type | Location, Time | Spec. range [nm] | Spec. resolution [nm] | Spectrograph (focal length) | $H_2O$ VMR % |
|---|---|---|---|---|---|---|
| ANT XXVIII | MAX-DOAS | Atlantic<br>54°N - 70°S<br>October - December 2011 | 277-413<br>390-617 | 0.7<br>0.9 | OMT<br>f=60 mm | 0.5–3.0 |
| M91 | MAX-DOAS | Peru, coastal upwelling<br>5°S 82°W–16°S 75° W<br>December 1st–25th 2012 | 324-467 | 0.45 | Acton 300i<br>f=300 mm | 1.6–2.4 |
| *HD15* | LP-DOAS | Heidelberg<br>49°25'N 8°43'W<br>August+September 2015 | 329–371<br>426–465 | 0.2<br>0.2 | Acton 300i<br>f=300 mm | 0.4–1.3 |

**Table 1.** Campaigns of which measurements were used. The cruise tracks of the ship-borne MAX-DOAS measurements are shown in Figure 2.

$I_0(\lambda) \cdot \exp(-\tau(\lambda))$. The optical density $\tau(\lambda)$ is calculated from a reference spectrum $I_0(\lambda)$ and a measurement spectrum $I(\lambda)$, $\tau(\lambda) = -\ln \frac{I(\lambda)}{I_0(\lambda)}$. To be independent of broad-band extinction by molecules and particles, the measured OD is partly compensated by a broad-band polynomial $p(\lambda)$ or filtered into a broad-band and narrow-band contribution. Characteristic and narrow-band absorption features of different absorbing trace-gas species with the cross-section $\sigma_i(\lambda)$ are then used to determine their respective concentrations $c_i(l)$ along the light path L:

$$\tau(\lambda) = \sum_i \sigma_i(\lambda) \int_0^L c_i(l)dl + p(\lambda) \tag{1}$$

The column density $S_i = \int_0^L c_i(l)dl$ is calculated by a fitting routine, which is applied to data from a given wavelength interval with a width of several nm to several 10 nm. The absorption path $L$ is known for LP-DOAS measurements and can be estimated or calculated from radiative transfer models for MAX-DOAS measurements. The high resolution literature cross-sections $\sigma_{L,i}$ are convoluted with the measured instrument function H of the respective setup to obtain $\sigma_i = H \otimes \sigma_{L,i}$, the absorption cross-section as it would be determined by the instrument. The instrument slit function is usually measured by observing individual atomic emission lines of mercury, which have a width which is two orders of magnitude smaller than the resolution of the instrument (Sansonetti et al., 1996).

LP-DOAS measurements (subsection 3.1) have the advantage of a well-defined light-path, but typically do not yield as small residuals as MAX-DOAS (subsection 3.2) observations. The disadvantage of MAX-DOAS measurements is, that their effective

light-path length depends on various factors such as atmospheric state (aerosols, clouds), which is often not known precisely. This needs to be explicitly considered in the data evaluation (subsection 4.2).

### 3.1 LP-DOAS Measurements

A description of the LP-DOAS instrument used here can be found in Pöhler et al. (2010) and Eger (2014). The total light path used for the measurements reported was 6.12 km long: Above the city of Heidelberg from the roof of the Institute of Environmental Physics to retro-reflectors mounted at the train station 'Molkenkur'. The spectral resolution was 0.2 nm in both spectral ranges.

The Longpath(LP)-DOAS instrument is based on an artificial light source (here a a LASER-driven light source Energetiq LDLS-EQ-99), retro reflectors, a telescope and a spectrometer. The light is sent by a telescope across the measurement distance to a retro reflector, which reflects the light back onto the same telescope. It collects the received light and transfers it to a spectrograph. A sequence of background measurements (i.e. measurements with the light source switched off or blocked), light-source spectrum measurements without absorption and actual measurement spectra is used to ensure independence of the measured spectra from external sunlight and instrumental instabilities (Pöhler et al., 2010). The LP-DOAS setup has the advantage that the actual light path is well-defined and thus average concentrations of absorbing molecules can be directly derived, also measurements at night are possible.

The optical density $\tau(\lambda)$ is calculated from a background corrected light source spectrum and a background corrected atmospheric spectrum and filtered by a binomial high-pass with 1000 iterations. The convoluted and high-pass filtered literature cross-sections listed in Table 2 are then fitted in the respective fitting interval to the corrected OD.

### 3.2 MAX-DOAS Measurements

Hönninger and Platt (2002) described the method of Multi-Axis DOAS (MAX-DOAS) measurements which improve the sensitivity of passive DOAS observations at altitude ranges close to the instrument (i.e. up to a few km). It uses scattered sunlight collected by a telescope pointing towards the sky at different elevation angles $\alpha$. The horizon is here defined as $\alpha = 0°$, zenith viewing direction as $\alpha = 90°$. Each elevation has a different sensitivity for absorptions in different heights of the atmosphere. Low elevation angles have a higher sensitivity to absorbers close to the surface, because the additional light path compared to a zenith spectrum recorded at the same time and location is mostly located within the lowermost layers of the atmosphere (Hönninger et al., 2004).

The slant column density (SCD) is defined as the integral over the concentration $\rho$ along the light path $L$ and is hence given in units of $molecules\ cm^{-2}$.

$$S = \int_L \rho(s)ds \tag{2}$$

From MAX-DOAS measurements differential slant column densities (dSCDs) can be calculated for each fitted trace gas: A Fraunhofer reference spectrum $I_0(\lambda)$ is chosen from one of the measurement spectra and the $\mathrm{dSCD}(\alpha) = \mathrm{SCD}(\alpha) - \mathrm{SCD}_{ref}$

| | | MAX-DOAS | | | | | LP-DOAS | | |
|---|---|---|---|---|---|---|---|---|---|
| | T [K] | $O_4$/$H_2O$ | $O_4$/$H_2O$ | HONO | BrO | OClO | $H_2O$ | $H_2O$ | |
| Wavelength interval [nm] | Start | 340 | 452 | 337 | 332 | 332 | 356 | 441 | |
| | End | 380 | 499 | 375 | 358 | 370 | 370 | 450 | |
| $H_2O$ vapour | 298 | | × | | | | | × | HITEMP (Rothman et al., 2010) |
| | | × | | × | × | × | × | | Polyansky et al. (2016) |
| $O_4$ | 293 | × | × | × | × | × | × | × | Thalman and Volkamer (2013) |
| | 273 | (×) | | | | | (×) | | |
| | 203 | (×) | | | | | | | |
| | 287 | (×) | | | | | | | Hermans et al. (2003) |
| | 296 | (×) | | | | | | | Greenblatt et al. (1990) |
| $O_3$ | 223 | × | × | × | × | × | | | Serdyuchenko et al. (2014) |
| | 243 | | | × | × | × | | | |
| | 293 | | | | | | × | | |
| HCHO | | × | | × | × | × | × | | Chance and Orphal (2011) |
| HONO | | | | × | | × | | | Stutz et al. (1999) |
| BrO | | × | | × | × | × | | | Fleischmann (2004) |
| OClO | | | | | | × | | | Bogumil et al. (2003) |
| $SO_2$ | | | | (×) | | | | | Vandaele et al. (2009) |
| $NO_2$ | 293 | × | × | × | × | × | (×) | | Vandaele et al. (1998) |
| $NO_2$ | 293 | | | | | | × | × | Voigt et al. (2001) |
| $NO_2$ absorption cell | 293K | | | | | | (×) | | |
| Ring Spectrum at | 273K | × | × | × | × | × | | | DOASIS (Kraus, 2006) |
| | 243K | × | | × | × | × | | | which uses Bussemer (1993) |
| Ring Spectrum $\cdot\lambda^4$ | | × | × | × | × | | | | Wagner et al. (2009) |
| Polynomial degree | | 3 | 3 | 5 | 3 | 4 | 3 | 3 | |
| Add. Polynomial degree | | 1 | 1 | 1 | 1 | 1 | 0 | 0 | |

**Table 2.** Retrieval wavelength intervals and reference spectra for the MAX-DOAS and LP-DOAS measurements. Literature cross-sections listed in brackets were used for sensitivity studies.

is obtained from the DOAS fit for each elevation angle $\alpha$ relative to the Fraunhofer reference. Typically a zenith spectrum is taken as reference and thus $\text{SCD}_{ref} = \text{SCD}(90°)$ In the measurements reported here, the DOAS fit includes the cross-sections listed in Table 2. By choosing references recorded shortly before and after the measurement spectrum the influence of the instrumental instabilities on the result was minimized as well as the influence of stratospheric absorbers.

#### 3.2.1 The MAX-DOAS instrument during *ANT XXVIII/1-2*

The MAX-DOAS instrument operated during Polarstern cruise *ANT XXVIII/1-2* consists of a telescope unit mounted on the deck of Polarstern at port-side, which actively corrects for the roll movement of the ship, and a spectrometer unit with two temperature stabilized OMT spectrometers (f=60 mm, $|\Delta T| < 0.1$°C, $\Delta\lambda < 0.01\,nm$), which had both been modified to minimize instrumental stray light (Lampel, 2014). Both spectrometers use back-thinned and peltier-cooled Hamamatsu S10141 CCD-detectors in order to have a high quantum efficiency in the UV range. The optical resolution of the instrument during this campaign was 0.7 nm and 0.9 nm and it covered a spectral range from 277–413 nm and 390–617 nm, respectively. Spectra were recorded for two minutes each at 7 elevation angles of 90° (zenith), 40, 20, 10, 5, 3, 1°, respectively, as long as solar zenith angles (SZA) were below $85°$. Glyoxal data from this campaign was published in Mahajan et al. (2014).

#### 3.2.2 The MAX-DOAS Instrument during M91

A description of the instrument operated during SOPRAN cruise M91 can be found in Großmann et al. (2013). The optical resolution of the instrument during this campaign was 0.45 nm and it covered a spectral range from 324 nm to 467 nm. The telescope elevation control unit actively compensated the ship's roll movement. Spectra were recorded for one minute each at 8 elevation angles of 90° (zenith), 40, 20, 10, 6, 4, 2, 1°, respectively, as long as solar zenith angles (SZA) were $\leq 85°$.

### 3.3 Spectral retrieval (MAX-DOAS)

The fit settings are summarized in Table 2, example fits are shown in Figure 5. As Fraunhofer reference spectra the sum of the two 40° spectra closest in time were used. Spectra recorded at a telescope elevation of 90° were not used as reference spectra, since they could have been influenced by direct sunlight during each of the MAX-DOAS campaigns close to the equator. The wavelength calibration was performed using recorded mercury discharge lamp spectra. On *ANT XXVIII/1-2* these were recorded automatically each night together with offset and dark-current spectra, during M91 they were recorded manually.

Measurement errors are calculated as twice the DOAS fit error, according to Stutz and Platt (1996). This estimate is justified, because the standard deviation of the residual of the linear fit shown in Figure 6 amounts to 2.1 times the average DOAS fit error and the residual spectra from the DOAS fit are dominated by noise in the UV.

For the water vapour absorption near 363nm, the wavelength interval was chosen using the technique described in Vogel et al. (2013) on spectra recorded on one individual day (November 15th, 2011 at about 6°N and 17°W) of the *ANT XXVIII/1-2* data set using the $O_4$ cross-section at 298K by Thalman and Volkamer (2013): For narrower wavelength ranges beginning above 345 nm and ending below 375 nm lower $H_2O$ dSCDs were observed during the day. However the standard deviations of the

$H_2O$ dSCDs for these retrieval intervals are with $5-6\times10^{23}$ molec cm$^{-2}$ (uncorrected) as large as the mean dSCDs. For the larger fit intervals the standard deviation is significantly smaller ($1-2\times10^{23}$ molec cm$^{-2}$) and the ratio of standard deviation of $H_2O$ dSCDs and the average fit error is close to 2, as expected from Stutz and Platt (1996). For the broader fit intervals the $H_2O$ dSCD varies for fit intervals within 330-390 nm with a standard deviation of 16% of mean $H_2O$ dSCD. We thus estimate the error due to the choice of fit settings to be below 20%. We assume that the small absorption structures of BrO and HCHO, which are not sufficiently constrained within fit intervals beginning above 345 nm cause this effect and/or possible compensation of the relatively broad $O_4$ absorption by the DOAS polynomial. When including HONO in the DOAS analysis for this day with low $NO_2$ concentrations and thus presumably low HONO concentrations, enhanced HONO and $H_2O$ dSCDs are observed simultaneously for fit intervals ending above 382 nm.

### 3.3.1 The blue spectral range

The effective center of the respective absorptions of $O_4$ and $H_2O$ can be calculated for each fit interval $[\lambda_1,\lambda_2]$ using

$$\lambda_m = \frac{1}{\int_{\lambda_1}^{\lambda_2}\sigma(\lambda)\mathrm{d}\lambda}\int_{\lambda_1}^{\lambda_2}\lambda\times\sigma(\lambda)\mathrm{d}\lambda \tag{3}$$

In the wavelength interval from 452-499 nm, the effective center of the water vapour absorptions of $\lambda_m^{H_2O}=479$ nm is close to the effective center of the $O_4$ absorptions at $\lambda_m^{O_4}=476$ nm.

The fit range was chosen to have similar effective centers of absorptions of $O_4$ and $H_2O$ in order to have comparable conditions for radiative transfer at both wavelengths.

HITEMP was chosen for the water vapour absorption cross-section in the blue wavelength region. The differences in the blue wavelength region to HITRAN 2012 are negligible at a spectral resolution of 0.5 nm. HITEMP was chosen instead of POKAZATEL in the blue wavelength range, as already a couple of previous publications use this cross-section in the blue wavelength range (see e.g. Lampel et al., 2015b, and references therein). As described in subsection 4.12 better agreement with observations was found for HITEMP than for POKAZATEL from 452-499 nm.

### 3.3.2 The near-UV spectral range

In the analyzed wavelength interval of 340-380 nm the absorption structures of $O_4$ and $H_2O$ are centered around $\lambda_m^{O_4}=361$ nm and $\lambda_m^{H_2O}=364$ nm.

As the observed optical depth (OD) in the fit ranges around 360 nm are small, except for the absorption of $O_4$ and the OD related to the Ring effect, it was necessary to include in addition to the normal Ring spectrum the temperature dependence of the Ring spectrum. This was calculated from the difference of Ring spectra $R(T)$ calculated at T=273 K and T=243 K using DOASIS: $\Delta R/\Delta T=(R(T-\Delta T)-R(T))/\Delta T$. The temperature dependence of the derivative of the Ring spectrum with respect to temperature is smaller than 0.5% / 1K, therefore it was sufficient to use one individual spectrum to linearise this effect. The OD associated with the Ring spectrum temperature dependence amounts to up to $5\times10^{-4}$ for the M91 data set when using a Ring spectrum calculated at 273K.

The contribution of vibrational Raman scattering of air on measurements in this spectral range could be correlated to the size of the Ring effect and agreed in its magnitude with the calculations given in Lampel et al. (2015a). Its effect on the results presented here was however neglible and was only consistently observed when co-adding more than 4 elevation sequences and for RMS of the resulting residuals of less than $1 \times 10^{-4}$. The effect of the wavelength dependence of the AMF for the $O_4$ absorptions at 344, 361 and 380 nm was found to be negligible for the spectral retrieval of water vapour absorption in this spectral range.

## 4 Results and Discussion

Starting with the largest absorption band below 380 nm listed in *POKAZATEL*, we show first experimental evidence of water vapour absorption in the UV from LP-DOAS measurements (subsection 4.1), which are complemented by even clearer detection of this absorption by MAX-DOAS observations (subsection 4.2). The magnitude of the absorption is quantified by comparison to water vapour absorption in the blue spectral range. From these results based on MAX-DOAS observations, a correction of the strength of the water vapour absorption band listed in *POKAZATEL* is derived.

We then also estimated the magnitude of the weaker water vapour absorption bands at 335nm (subsection 4.6) and 373 nm (subsection 4.7).

### 4.1 LP-DOAS: Detection of water vapour absorption at 363 nm

Measurements between 22 August and 24 September, 2015 were used for this analysis. Measurement spectra were co-added in order to reduce the RMS of the residual in the UV fit interval to values of $1.5 \pm 0.3 \times 10^{-4}$ along the total light path of 6.12 km, which resulted in a time resolution of two hours. This corresponds to an exposure time of about 15 minutes for each measurement spectrum. Due to need to change the wavelength setting of the spectrometer between the different spectral windows around 440nm and 360nm, the time for each measurement sequence is shorter than the total time resolution.

A weak correlation of the water vapour absorption around 363 nm to the absorption at 442 nm was found with a correlation coefficient of $R^2 = 0.25$ (Figure 4) for individual measurements. The rather weak correlation is due to the large individual measurement errors, which can be directly seen by the large variations from one measurement to the next. For daily averaged values the correlation amounts to $R^2 = 0.61$. Further co-adding of spectral measurement data could not reduce the measurement errors further, as systematic residual structures appear (see Figure 3). Furthermore large $NO_2$ concentrations of up to 20 ppb led to additional residual structures. Selecting measurement spectra according to the $NO_2$ concentration or RMS did not improve the correlation.

As the measurement period was in late summer with temperatures between 9–36°C and relative humidity between 20–96% leading to a water vapour VMR between 0.4-1.3% (5–16.5 g m$^{-3}$), low as well as high VMRs are not well represented in this data set. This increases the error in the correlation of water vapour column densities determined in both wavelength intervals (see Table 2). Linear regression yields a relative magnitude of the absorption near 363 nm of $2.31 \pm 0.25$ and an offset of $1.6 \pm 4.5 \times 10^{22}$ molec cm$^{-2}$ . Fixing the offset to zero yields a scaling factor for the absorption cross-section near 363 nm

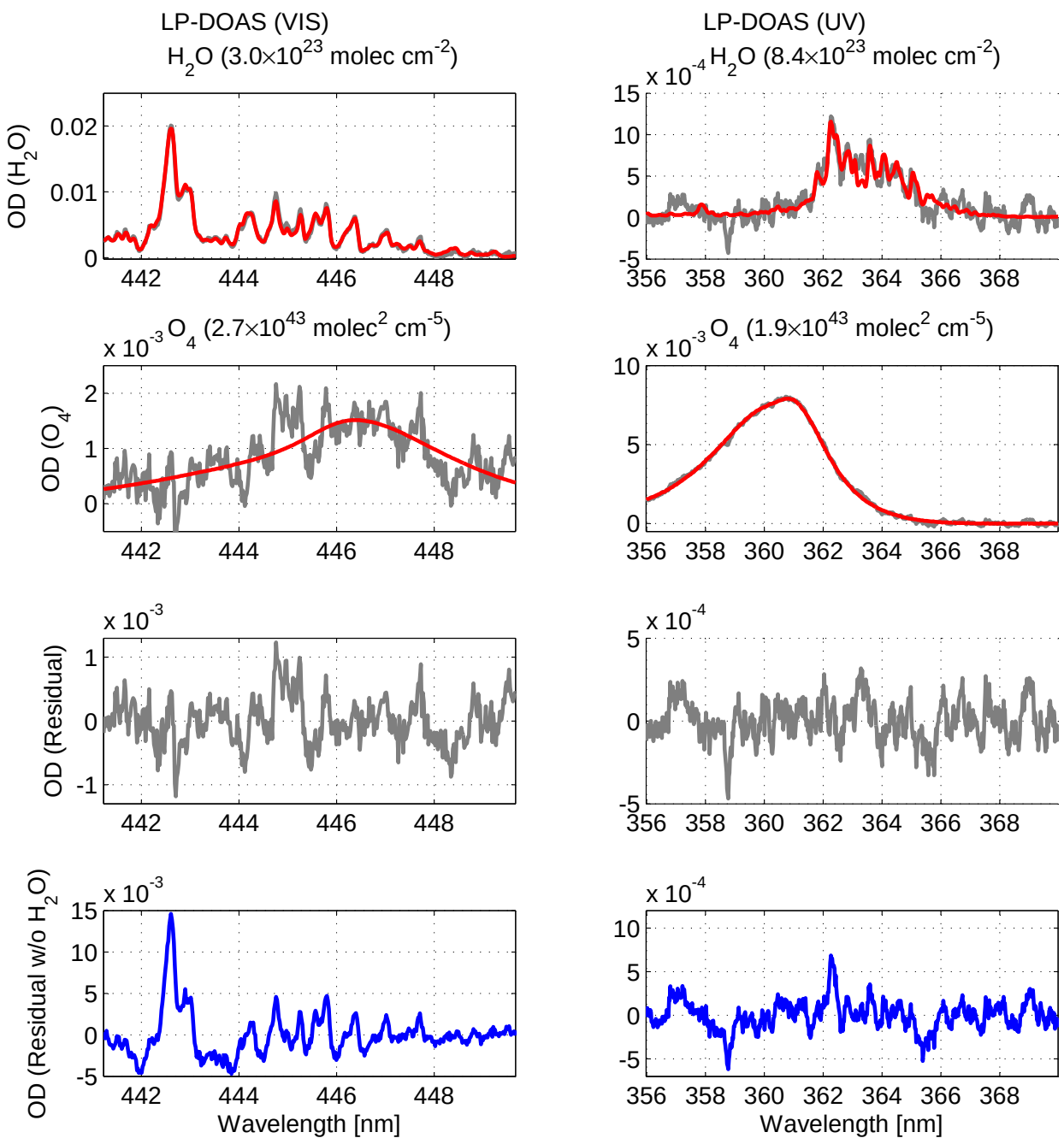

**Figure 3.** A LP-DOAS fit result for the fitting intervals around 363 and 442 nm. The spectra were recorded on August 29th, 2015 between 20:58 and 21:45 UTC. Top left panel: At 442 nm the $H_2O$ dSCD $(3.0 \pm 0.04) \times 10^{23}$ molec cm$^{-2}$ ($O_4$ dSCD $(2.7 \pm 0.6) \times 10^{43}$ molec$^2$ cm$^{-5}$ ). Top right panel: At 360 nm the $H_2O$ dSCD $(8.4 \pm 0.6) \times 10^{23}$ molec cm$^{-2}$ ($O_4$ dSCD $(1.85 \pm 0.03) \times 10^{43}$ molec$^2$ cm$^{-5}$ ).

of $2.39 \pm 0.05$. This means, the *POKAZATEL* line lists underestimates the observed absorptions near 363 nm by a factor of $2.39 \pm 0.05$. The measurement error will contribute significantly to the error of the scaling factor, as it is about 30% of the maximally measured column density near 363 nm. Thus we estimate the overall scaling factor from LP-DOAS measurements to be $2.4 \pm 0.7$.

### 4.2 MAX-DOAS: Detection of water vapour absorption near 363 nm

The absorption of water vapour was detected at about 363 nm (27548 cm$^{-1}$) in measurements from *ANT XXVIII/1-2* and M91, using a fit interval from 340-380 nm (26316-27548 cm$^{-1}$) according to Table 2. The maximum signal-to noise ratio during both cruises (ratio between fitted $H_2O$ dSCD and measurement error) were 14 and 10, respectively (15 and 20, respectively, for 16 co-added elevation sequences). The corresponding dSCD values showed the typical separation for each elevation angle as observed for water vapour absorptions in the blue wavelength range. The corresponding spectra are shown in Figure 5.

The retrieved water vapour dSCDs at 363 nm were compared to the 20-times stronger water vapour absorptions between 452-499 nm (20040–22124 cm$^{-1}$) for the *ANT XXVIII/1-2* dataset. To correct for possible influences of varying radiative transfer conditions (which may result in different light path lengths and thus different dSCDs), the $H_2O$ dSCDs retrieved from

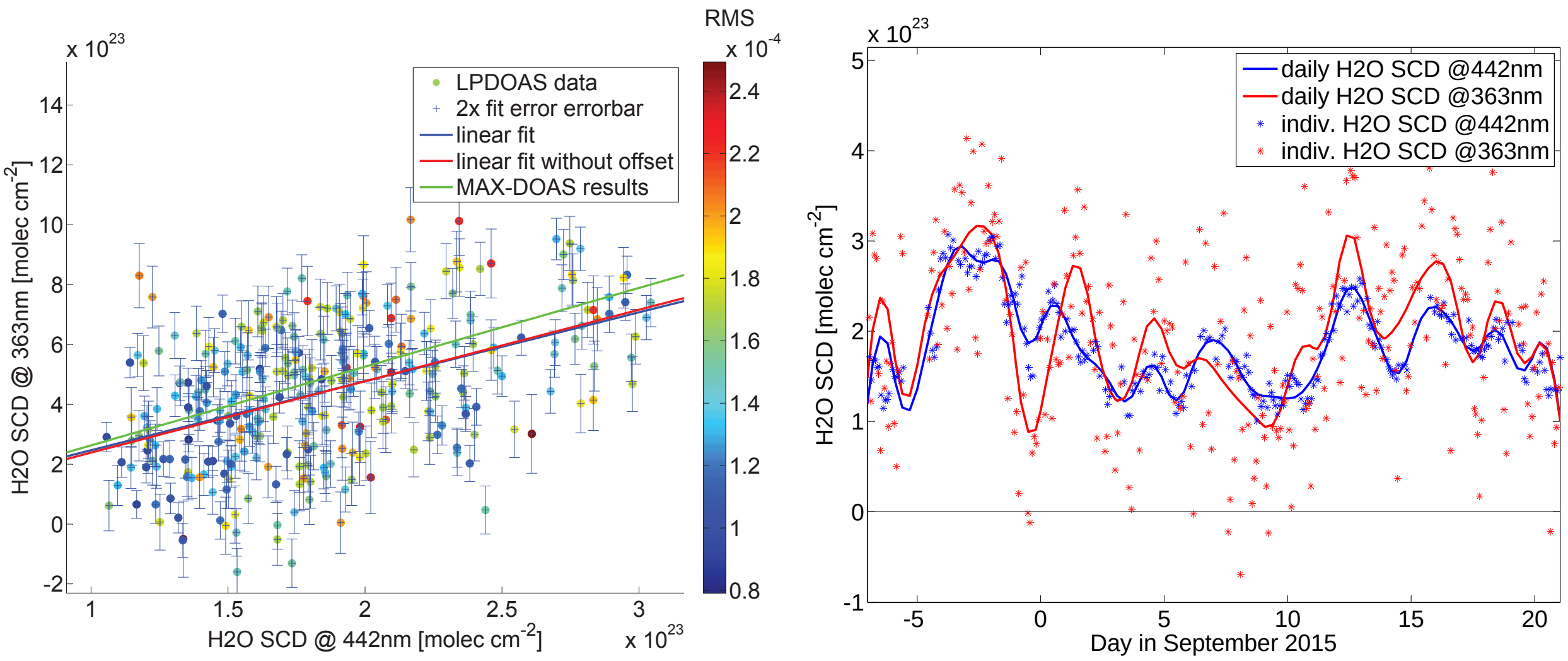

**Figure 4. Left:** Correlation of $H_2O$ SCDs from LP-DOAS measurements near 363 and 442 nm. Also shown is the result from Table 3 line (1) from MAX-DOAS observations. **Right:** Time series of $H_2O$ column densities from LP-DOAS measurements near 363 and 442 nm. Values near 363 nm were corrected by the scaling factor $2.31 \pm 0.25$ determined from the correlation plot on the left.

both spectral windows were divided by the respective $O_4$ dSCD from the same fitting window. These fitting intervals were selected such according to Table 2, that the wavelength of the main absorptions of $O_4$ and $H_2O$ are at the same wavelength range in order to have approximately the same radiative transfer for both absorbers (see subsubsection 3.3.1). The absorption of $O_4$ is an indicator for the light path length, since the $O_4$ concentration is proportional to the square of the concentration of molecular oxygen, which has a well-defined and sufficiently constant concentration profile.

The scale height of $O_4$ is 4 km, the scale height of water vapour is typically 2 km (Wagner et al., 2013). MAX-DOAS measurements of trace gas dSCDs are most sensitive to the lowermost 2 km (e.g. Frieß et al., 2006). Thus for a given surface volume mixing ratio of water vapour, an almost constant ratio of $H_2O$ and $O_4$ dSCD is expected. Figure 6 shows that this approximation is valid for the ANT XXVIII measurements, as the correlation coefficients $R^2$ for the individual $O_4$ and $H_2O$ dSCDs are smaller (0.83 and 0.77) than the correlation coefficient $R^2 = 0.89$ for their ratio.

However, the different profile shapes can introduce deviations, which were investigated by radiative transfer modelling using the Monte Carlo radiative transfer model McArtim (Deutschmann et al., 2011). Assuming different water vapour surface concentrations (0.1–3%), water vapour scale heights of 1,2,3 km, an aerosol layer with an extinction of 0, 0.2, 1, 2, 10 $\text{km}^{-1}$ with a thickness of 1 and 3 km in an altitude of 0,1,2,3 km, the resulting simulated $H_2O/O_4$ dSCD ratios correlate for both wavelengths 363 nm and 477 nm with an $R^2 = 0.98$ and a slope of $1.00 \pm 0.02$. The intercept was fixed to zero. Elevation angles were $3, 5, 90°$. 6480 individual simulations were performed. A significant systematic dependence of the ratios on ground albedo, solar zenith angle and relative azimuth angle was not observed, each of them resulting in less than 1% change of the

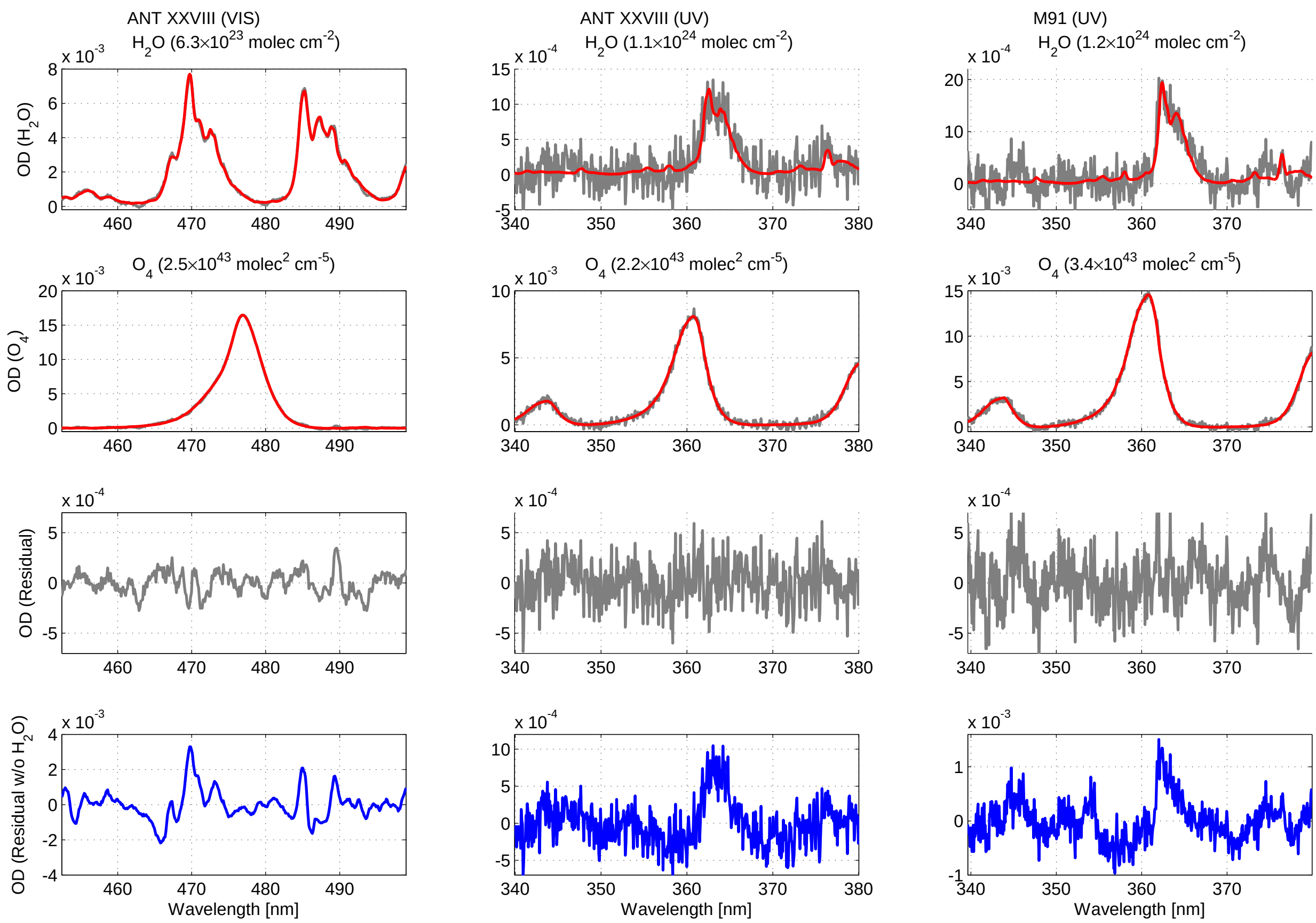

**Figure 5.** Fit results from *ANT XXVIII/1-2* and M91 showing the detection of water vapour absorptions at 477 nm and 363 nm; in red, the modeled absorptions according to the cross-sections listed in Table 2, in grey, the measured values. In blue, the residual is shown if no water vapour absorption was included in the fit. The fits from *ANT XXVIII/1-2* use a spectrum (exposure time: 120 s, spectral resolution 0.7 nm) from November 16th, 2011 at 13:20 UTC at 3°59′06″N 14°44′40″W at a telescope elevation angle of 3°. At 477 nm the $O_4$ dSCD is $(2.47 \pm 0.01) \times 10^{43}$ molec$^2$ cm$^{-5}$ , the $H_2O$ dSCD $(6.27 \pm 0.06) \times 10^{23}$ molec cm$^{-2}$ . At 360 nm the $O_4$ dSCD is $(2.18 \pm 0.04) \times 10^{43}$ molec$^2$ cm$^{-5}$ , the $H_2O$ dSCD $(1.13 \pm 0.16) \times 10^{24}$ molec cm$^{-2}$ . The fit from M91 is using one spectrum (exposure time: 60 s, spectral resolution 0.45 nm) recorded on December, 5th 2012, 19:44 UTC at 7°24′29″S 81°30′18″W at a telescope elevation of 3°. It shows an $O_4$ dSCD of $(3.43 \pm 0.02) \times 10^{43}$ molec$^2$ cm$^{-5}$ and a $H_2O$ dSCD of $(1.18 \pm 0.16) \times 10^{24}$ molec cm$^{-2}$ . All fits used the $O_4$ cross-section by Thalman and Volkamer (2013).

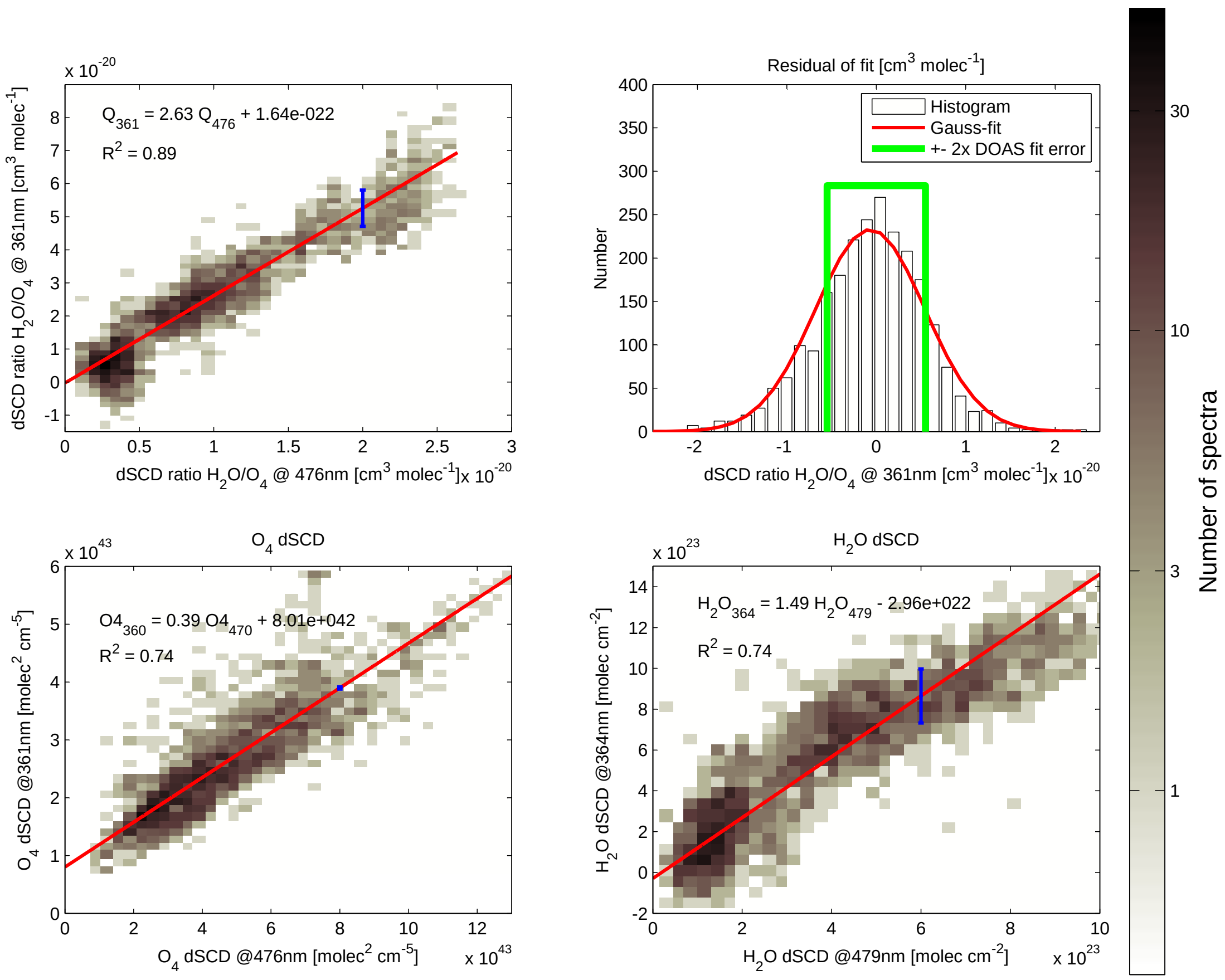

**Figure 6. Top left panel**: Ratio of water vapour dSCD and $O_4$ dSCD at 363 nm and 477 nm for a telescope elevation angle of 3 and 5° during *ANT XXVIII/1-2* using the $O_4$ cross-section by Thalman and Volkamer (2013). Error bars are calculated from fit errors of both absorbers. Error bars for the ratios at 477 nm are omitted. They are more than one order of magnitude smaller than those at 360 nm. A ratio of $10^{-20}$ cm$^3$ molec$^{-1}$ corresponds to an absolute water vapour mixing ratio of 0.01 at ground-level or a vertical column density of $5 \cdot 10^{22}$ molec cm$^{-2}$ or 15 kg $H_2O$ m$^{-2}$ assuming a scale height of 2 km. **Top right panel**: The residual of the linear fit shows a Gaussian distribution and agrees with respect to its width of $\sigma = 6.12 \times 10^{-21}$ cm$^3$ molec$^{-1}$ with the mean measurement error (two times DOAS fit error, $2.75 \pm 0.92 \times 10^{-21}$ cm$^3$ molec$^{-1}$ ) obtained from the DOAS fit. The individual correlations of $H_2O$ and $O_4$ dSCDs are shown in the **lower panels**, which show individually smaller correlation coefficients than their respective ratios at 360 and 477 nm.

| | Type | $O_4$ cross-section | $R^2$ | Slope | Syst. Error [%] | Offset [$cm^3$ $molec^{-1}$] | $n$ |
|---|---|---|---|---|---|---|---|
| 1 | MAX-DOAS | Thalman 273K | 0.89 | 2.63(1) | 8 | $0.16(4) \times 10^{-21}$ | 2621 |
| 2 | MAX-DOAS | Thalman 273K free shift | 0.88 | 2.61(1) | 8 | $0.34(4) \times 10^{-21}$ | 2634 |
| 3 | MAX-DOAS | Thalman 273K+293K | 0.83 | 2.39(1) | 8 | $7.25(5) \times 10^{-21}$ | 2562 |
| 4 | MAX-DOAS | Hermans | 0.86 | 2.62(1) | 8 | $4.22(4) \times 10^{-21}$ | 2630 |
| 5 | MAX-DOAS | Greenblatt | 0.84 | 2.55(1) | 9 | $21.1(1) \times 10^{-21}$ | 2183 |
| 6 | MAX-DOAS | Greenblatt (shifted by 0.2 nm) | 0.89 | 2.58(1) | 11 | $10.1(1) \times 10^{-21}$ | 2586 |
| 7 | LP-DOAS | Thalman 293K | 0.25 | 2.31(25) | 30 | $1(3) \times 10^{-21}$ | 320 |

**Table 3.** Results from Figure 6 to determine the relative magnitude of the water vapour absorption at 363 nm compared to 477 nm using the HITEMP cross-section for different retrieval settings using different $O_4$ cross-sections. Values in brackets denote the error of the last digits of the respective value calculated from the error-weighted linear regression. For LP-DOAS measurements (see subsection 4.1) the correlation was done for SCDs instead of $H_2O/O_4$ dSCD ratios, because the light path was constant. The offset (LP-DOAS) was however normalized by the mean O4 dSCD at 360 nm in order to have comparable values. The systematic error of the slope was determined by using the typical relative measurement error of water vapour for measurements at a dSCD of $3 \times 10^{23}$ molec $cm^{-2}$ determined in the respective blue wavelength range.

simulated $O_4/H_2O$ ratio. Simulations with small $O_4$ dSCDs, which result in a large simulation error for the $H_2O/O_4$ dSCD ratio, were removed analogously to the measurements.

The Ångström exponent was varied using values of 0.0, 0.5 and 1.0 according to AERONET AOD measurements during *ANT XXVIII/1* (Smirnov et al., 2009)[1]. The effect on the ratio was however also smaller than 1%.

As for the measured data, the correlation of the simulated $O_4$ or $H_2O$ dSCDs individually is significantly worse with $R^2_{O_4} = 0.74$ and $R^2_{H_2O} = 0.91$. The slope of a linear polynomial fit to the $O_4$ dSCDs at 360 nm and 470 nm is similar to the observed values.

For Figure 6, measurements at an elevation angle of 3-5° with an RMS of less than $8 \times 10^{-4}$ (UV) and $4 \times 10^{-4}$ (VIS) were used, additionally the error of the $H_2O/O_4$ ratio calculated from the fit errors of both trace gases had to be below $5 \times 10^{-21}$ $cm^3$ $molec^{-1}$ (UV) and $3 \times 10^{-22}$ $cm^3$ $molec^{-1}$ (VIS). This implicitly removes all measurements with low $O_4$ dSCDs, which is the case for fog and very low clouds.

As seen from Figure 6, the $H_2O/O_4$ dSCD ratios from *ANT XXVIII/1-2* correlate well for the wavelength ranges around 360 nm and around 477 nm with an $R^2 = 0.89$. However, the absolute magnitude of the absorption cross-section near 363nm is underestimated by a factor of $2.6 \pm 0.3$ (see Table 3).

The $O_4$ cross-section is known to change its shape with changing temperature (Pfeilsticker et al., 2001; Thalman and Volkamer, 2013). As this effect could potentially introduce similar latitudinal dependencies as the water vapour distribution, the

[1]http://aeronet.gsfc.nasa.gov/new_web/cruises_new/Polarstern_Fall_11.html

spectral analysis was run in addition to the original analysis including two $O_4$ cross-sections at 293K and 273K. This changed the slope of the correlation shown in Figure 6 by -10% from 2.63 to 2.39 (see Table 3). In addition, an increase is observed for the offset of the linear fit, which should be ideally zero. Fixing the linear regression line for high water vapour content at the observed values, this increase in the offset of the linear fit corresponds to the observed change in the slope. We therefore conclude that the observed absorption structure is not caused by the temperature dependence of the $O_4$ absorption cross-section, but indeed by water vapour absorption, as this offset is observed in polar regions, where almost no water vapour absorption is expected. Note that this offset is still small and amounts to 10 % ($7.25 \times 10^{-21}$ cm$^3$ molec$^{-1}$) of the observed maximum ratio of $H_2O/O_4$ dSCDs shown in Table 3.

A spectral shift of the $O_4$ literature cross-section can effectively compensate parts of the water vapour absorption cross-section at 363 nm. This is discussed in subsection 4.5. However stable results were even obtained when the shift of the set of literature cross-sections was determined by the Levenberg-Marquardt algorithm of the DOAS fit, as shown in row (2) in Table 3.

As seen from Table 3 the resulting slopes from Figure 6 agree within their respective errors for different $O_4$ cross-sections. The $O_4$ absorption by Greenblatt et al. (1990) shows a systematic shift for the absorption at 360 nm and was therefore analysed once with the original wavelength calibration and once shifted by 0.2 nm (used e.g. in Pinardi et al. (2013)). The results of the shifted $O_4$ cross-section include more measurements but still show a significant offset of the linear regression. The results using the Hermans et al. (1999) $O_4$ cross-section seem more reliable, as more data points can be used and the offset of the slope is smaller. The most consistent results are obtained when using the $O_4$ cross-section by Thalman and Volkamer (2013), showing a small offset and the highest correlation coefficient.

### 4.3 Differences using different dipole moment surfaces (DMS)

The *POKAZATEL* line list employs the DMS from Lodi et al. (2011), while the *POKAZATEL (CVR)* line list employs the DMS by Lodi et al. (2008), while using the same PES. This leads to significant differences in the intensities of the resulting line lists in the near-UV spectral region. The magnitude of the absorption between 362–365 nm in *POKAZATEL (CVR)* is on average 2.9 (ranges between 2.3–4.6) times larger than in *POKAZATEL*, and might therefore explain the observed discrepancy in the magnitude of the cross-section shown in subsection 4.2. However, the shape of the absorption band in the atmospheric measurements is significantly better predicted by *POKAZATEL*. Fitting *POKAZATEL (CVR)* to measured spectra from M91 leads to 20% higher RMS of the residual (see Figure 7) at low elevation angles. The additional absorption structures around 354 nm listed in *POKAZATEL (CVR)* are not found in observations (compare also Figure 10). These findings are consistent with the spectral analysis of data from *ANT XXVIII/1-2* .

*POKAZATEL (CVR)* also predicts water vapour absorption between 330-360 nm, which should be above our detection limit. These could however not be identified for either of the two line lists (see also subsection 4.6).

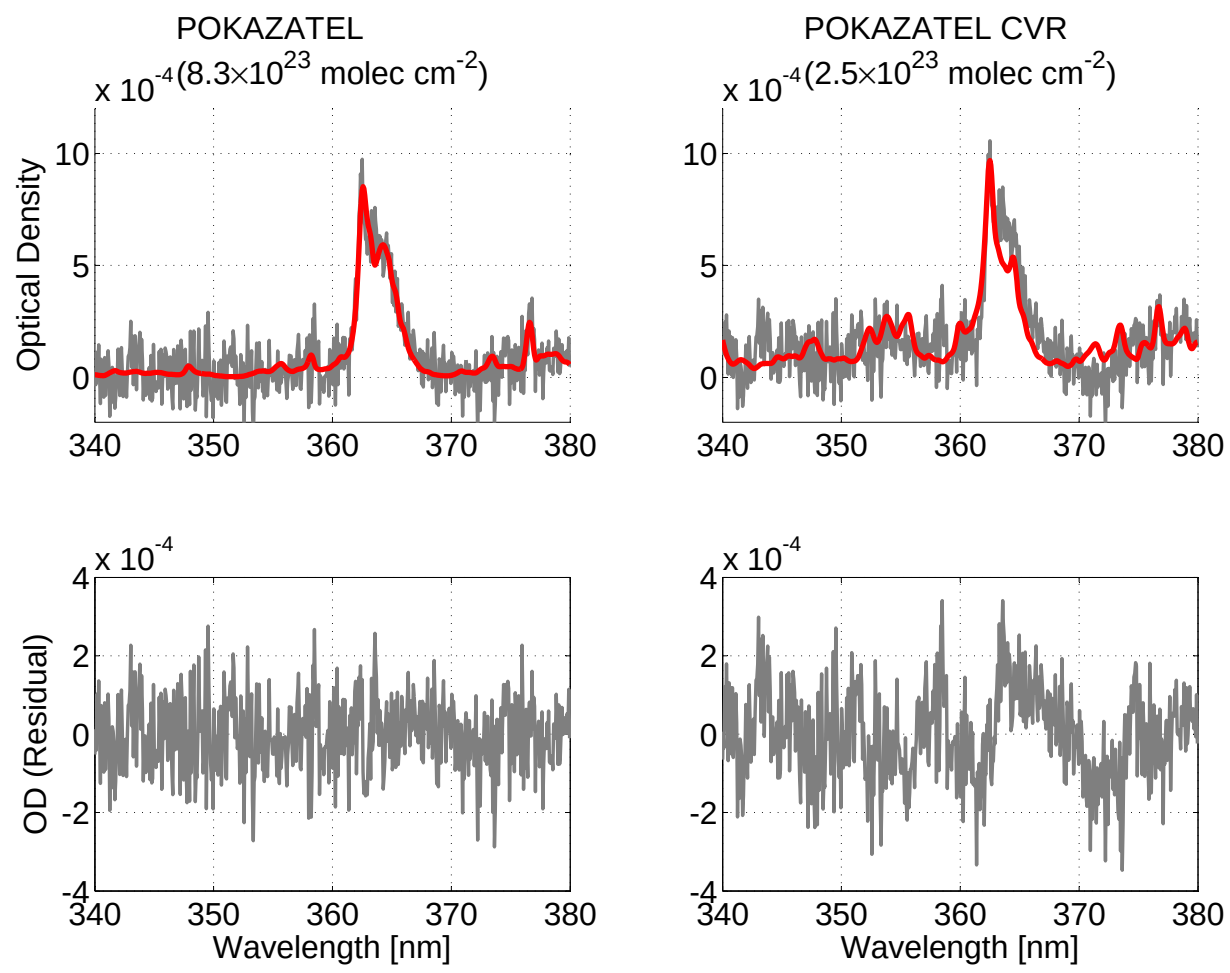

**Figure 7.** Two MAX-DOAS fits of the same measurement spectrum from M91 showing the detection of water vapour absorptions at 363 nm using two different DMS(see subsection 4.3). In order to reduce residual noise, the fit is using four spectra with a total exposure time of 240 s recorded on December, 22nd 2012, starting at 17:59 UTC at $15°31'$S $75°36'$W at a telescope elevation of $3°$. The *POKAZATEL (CVR)* line list shows a 20% larger residual than *POKAZATEL*, whose shape fits the observed optical density better.

### 4.4 Comparison to other line lists

As shown in Figure 1, other water vapour line lists also contain lines in the spectral range below 390 nm, which should be theoretically above typical detection limits of our measurements (often better than $10^{-4}$ along a light path of 10km). However in this spectral range BT2 and HITEMP are based on calculations only and have not yet been confirmed by laboratory or atmospheric measurements. The absorption at 380 nm should be clearly above the detection limit of the instrument used during M91, but as reported in Lampel et al. (2015b), it was not unambiguously found and showed inconsistencies. These two line lists show further absorption lines between 330–360 nm, which could also not be identified in Lampel et al. (2015b).

Fitting simultaneously a cross-section based on *POKAZATEL* and a cross-section based on HITEMP or BT2 to the measurements (M91), the optical density (from 340–380 nm) attributed to BT2 and HITEMP remained below $(3\pm12)\%$ and $(2\pm8)\%$, respectively, of the optical density of the water vapour absorption attributed to the *POKAZATEL* cross-section. The optical density attributed to BT2 and HITEMP was $(-1\pm6)\times10^{-5}$ and $(-1\pm4)\times10^{-5}$, respectively, while the *POKAZATEL* cross-section showed absorptions of $(4.5\pm4.3)\times10^{-4}$ for all spectra at all elevation angles of the M91 dataset with a RMS of the residual of less than $4\times10^{-4}$.

These findings demonstrate that the shape of the water vapour absorption appears to be better predicted in the *POKAZATEL* line list than in either the BT2 and the HITEMP. For HITEMP this was expected, since HITEMP is partly based on BT2, but a large number of individual lines had been removed due to the individual line intensity cut-off of $10^{-27}$ cm molec$^{-1}$ for these wavelengths (Rothman et al., 2010). This procedure leads to changes in absorption band shape and the significantly smaller water vapour absorption cross-section in HITEMP compared to BT2 as shown in Figure 1.

### 4.5 Compensation of $H_2O$ absorption by $O_4$ absorption near 363 nm

Since the water vapour absorption is found at the red flank of the $O_4$ absorption band at 361 nm, the absorption can be partly compensated by shifting the $O_4$ absorption band towards longer wavelengths. This effect is more clearly observed for the ANT XXVIII data-set than for the M91 data-set, due to the lower spectral resolution, which seems to match better the widths of the spectral absorption structures of $O_4$.

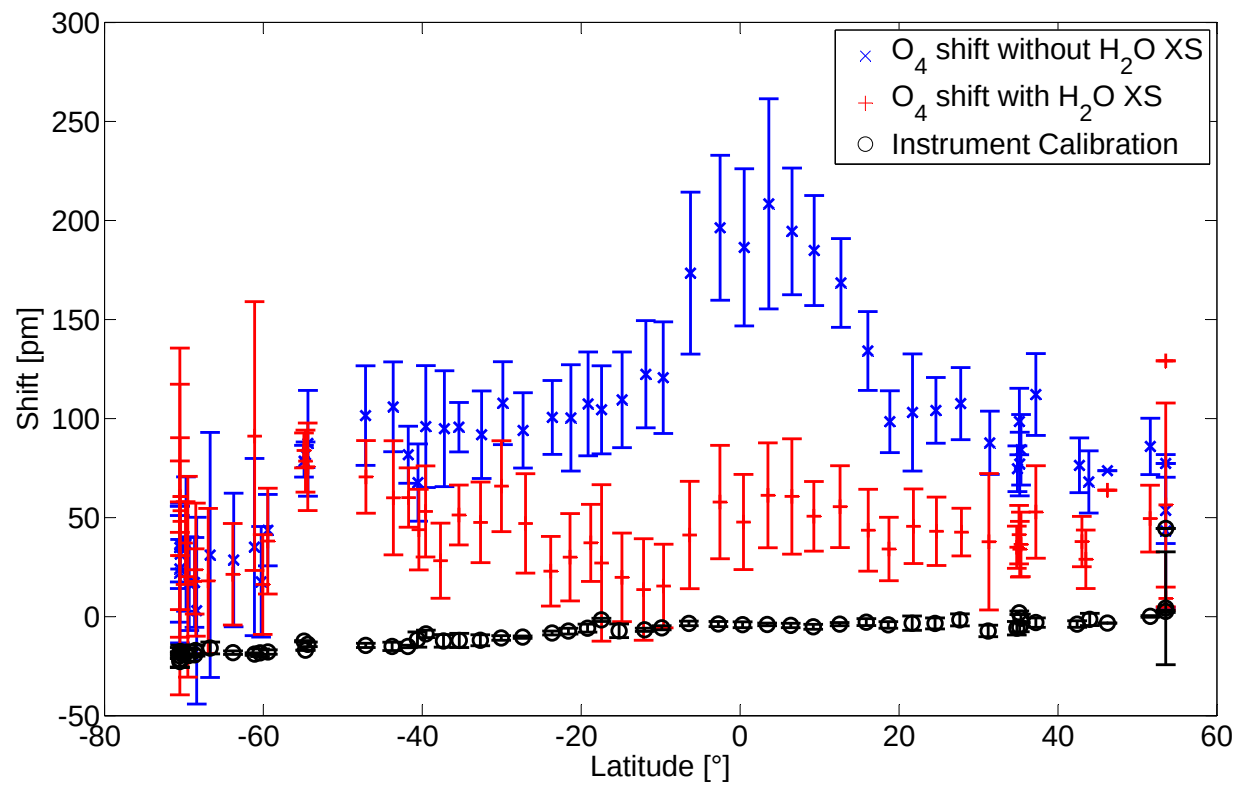

**Figure 8.** Error weighted daily averaged DOAS fit results for the shift of the $O_4$ cross-section for measurements with a signal-to-noise ratio for the $O_4$ dSCD of more than 50. For this evaluation, the shift of the $O_4$ cross-section was freely determined by the DOAS fit and not linked to the other absorption cross-sections. Error bars denote the standard deviation during one day. The shift of the instrumental calibration was determined from fit of the measured spectra to data from a convolved solar atlas.

When evaluating the *ANT XXVIII/1-2* data-set using the same settings as listed above in Table 2, but allowing for a spectral shift of the $O_4$ cross-section by Thalman and Volkamer (2013), a systematic shift of the $O_4$ cross-section of up to 0.20 nm relative to a Fraunhofer reference calibrated using the solar atlas of Chance and Kurucz (2010) is observed in tropical regions (shown in Figure 8). A systematic shift of the $O_4$ cross-section of up to 0.15 nm relative to a freely shifting $O_4$ cross-section from a fit including the *POKAZATEL* water vapour absorption cross-section is observed. When the water vapour absorption is included, the free shift of the $O_4$ cross-section shows a standard deviation of 0.035 nm for measurements with a signal-to-noise ratio of more than 50 for the $O_4$ dSCD. The instrument calibration shows a standard deviation of 0.007 nm due to a slow drift of 0.3 pm $d^{-1}$.

It was found that a small shift of $O_4$ with temperature (e.g. 0.05 nm as in Thalman and Volkamer (2013) from 273K–293K) cannot explain the apparent shift of the $O_4$ absorption when not considering the water vapour absorption.

As described in Beirle et al. (2013), a spectral shift can be linearised for small shifts by the derivative of the absorption cross-section with respect to wavelength using Taylor expansion. Turning the argument around, therefore a correlation of the size of the absorption structure of water vapour and the product of $O_4$ absorption and spectral shift (from a DOAS fit where water vapour absorptions are not considered) is expected. This correlation is found for *ANT XXVIII/1-2* data with $R^2 = 0.89$ and

a slope of $a_S = 6.78 \times 10^{18}$ nm molec cm$^{-3}$. For this instrument with a spectral resolution of 0.7 nm it thus means effectively that a water vapour dSCD of $S_{H_2O} = 5 \times 10^{23}$ molec cm$^{-2}$ and an $O_4$ dSCD of $S_{O_4} = 2.5 \times 10^{43}$ molec$^2$ cm$^{-5}$ leads to a shift of the $O_4$ cross-section by $a_S \cdot S_{H_2O}/S_{O_4} = 0.14 nm$, which was indeed observed in tropical regions as shown in Figure 8. The change in overall $O_4$ dSCD is discussed in subsection 4.11.

### 4.6 Upper limit for water vapour absorption at 335 nm

The water vapour absorption band at 335 nm in the *POKAZATEL* line list would amount to an OD of $1.2 \times 10^{-4}$ for a water vapour dSCD of $4 \times 10^{23}$ molec cm$^{-2}$ at a spectral resolution of 0.5 nm.

Analogous to the procedure described in subsection 4.2, the water vapour absorption band at 335 nm (fit range 332–358 nm) was compared to the water vapour absorption within the interval from 452-499 nm for the *ANT XXVIII/1-2* measurements, divided by the dSCD of the respective $O_4$ absorption band. A clear correlation was not observed ($R^2 < 0.2$) due to too large fit errors to detect water vapour in the BrO/HCHO fit range (fit settings: Table 2). The water vapour dSCD (at 335 nm) stayed below the average detection limit of $7 \times 10^{23}$ molec cm$^{-2}$ .

For the M91 MAX-DOAS measurements the detection limit was reduced by co-adding 16 elevation sequences. However, the correlation of water vapour dSCDs at 335 nm and 442 nm was small ($R^2 = 0.2$) and the $2\sigma$ detection limit of $6.5 \times 10^{23}$ molec cm$^{-2}$ was only exceeded for 10% of all spectra.

We therefore conclude that the predicted magnitude of the absorption at 335 nm is correct or overestimated, as we could not find it in our MAX-DOAS observations: If the shape of the water vapour absorption is correctly predicted by *POKAZATEL*, the magnitude of the differential water vapour cross-section from 332–358 nm at a spectral resolution of 0.45 nm-0.70 nm is smaller than $2.5 \times 10^{-28}$ cm$^2$ molec $^{-1}$.

### 4.7 Water vapour absorption around 376 nm

The literature values for the water vapour absorption cross-sections based on *POKAZATEL* and BT2 (and thus also HITEMP) differ by about one order of magnitude in the spectral region between 370 and 380 nm (compare Figure 1). Using the M91 MAX-DOAS measurements the absorptions listed in BT2 could not be unambiguously identified or its predicted absorption shape did not match the observed absorptions. We therefore apply here the *POKAZATEL* line list on data from the M91 campaign.

We use a fit range from 370-386 nm and the settings for the water vapour absorption at 363 nm without considering the water cross-section of $O_3$, HONO, BrO and HCHO. Co-added spectra based on four elevation sequences were used in order to reduce the average fit error to $2 \times 10^{23}$ molec cm$^{-2}$ (average RMS of the residual: $1.1 \times 10^{-4}$). The water vapour dSCD was compared to the water vapour dSCD from 340-380 nm, retrieved in Figure 4.2. The resulting correlation of dSCDs at 363 nm and 376 nm is significant with $R^2$=0.6 and a slope of $1.2 \pm 0.3$. A DOAS fit result is shown in Figure 9. As both absorption bands are at similar wavelengths and the absorptions are small, the difference in expected dSCDs introduced by differences in radiative transfer are negligible compared to the measurement error itself.

This shows that the water vapour absorptions at 376 nm is found in MAX-DOAS measurements and its magnitude is predicted in agreement with the absorption at 363 nm. It underestimates the absorption inferred from measurements by a factor of $3.1 \pm 0.7$.

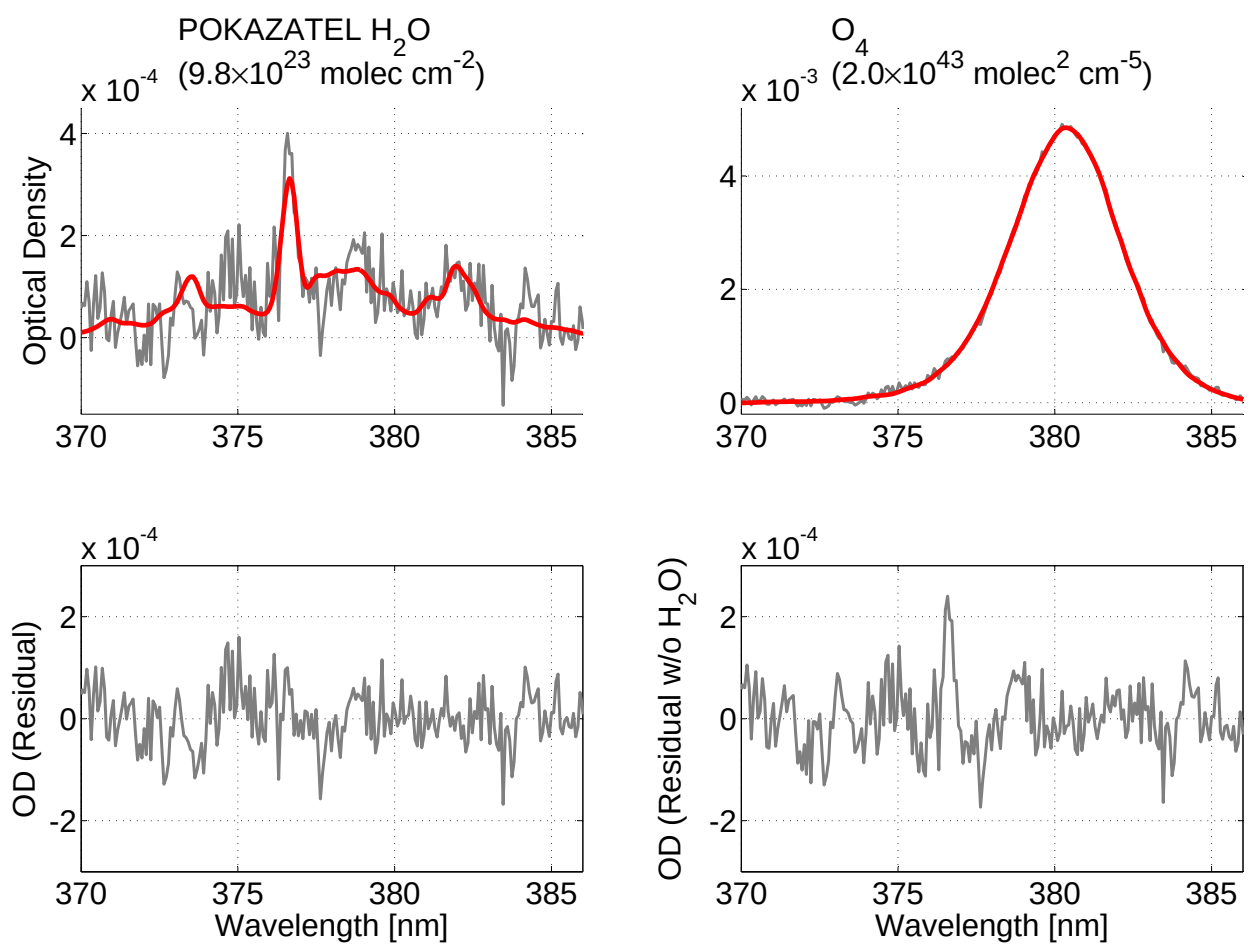

**Figure 9.** Fit result for the same MAX-DOAS spectrum as used in Figure 7 to show the water vapour absorption at 376 nm, which correlates for the M91 dataset with $R^2$=0.6 and a slope of $1.2\pm0.3$ with the water vapour absorption at 363 nm. The measurement error of this individual fit amounts to 20%.

### 4.8 Water vapour absorption below 330 nm

Du et al. (2013) reported significant water vapour absorptions of up to $2.94 \times 10^{-24}$ cm$^2$ molec$^{-1}$ at 330 nm and up to $2.19 \times 10^{-24}$ cm$^2$ molec$^{-1}$ at around 315 nm. Lampel et al. (2015b) could not confirm these findings and found upper limits for the differential absorption of water vapour from 332–370 nm of $3 \times 10^{-27}$ cm$^2$ molec$^{-1}$ from the M91 dataset which are two orders of magnitude smaller. Wilson et al. (2016) also could not confirm the values published by Du et al. (2013) between 325–420 nm. They estimated the water vapour absorption cross-section at a spectral resolution of 0.5 nm to be less than $2 \times 10^{-25}$ cm$^2$ molec$^{-1}$.

For a water vapour dSCD of $4 \times 10^{23}$ molec cm$^{-2}$ , the findings of Du et al. (2013) would result in differential optical depths around unity, which is unrealistic judging from observations of BrO and HCHO in the troposphere in wavelength intervals within 330–360 nm. The instrument operated during *ANT XXVIII/1-2* covers a wider spectral range than in Lampel et al. (2015b), we therefore applied the BrO/HCHO fit settings from Table 2 to a fit interval from 310–350 nm. The water vapour dSCD was obtained from the absorption listed in *POKAZATEL* at 364 nm divided by 2.63 as shown in Figure 4.2 to estimate the expected dSCD in this fit interval. A polynomial of degree 0–2 was applied in the fit in order to account broad-band absorptions and scattering. The resulting peak-to-peak (ptp) magnitudes of the residual are listed for an example measurement spectrum at 3° in Table 4. To avoid unnecessary compensation of potential water vapour absorption by other absorbers, their

| Polynomial degree | ptp residual | upper limit diff. $H_2O$ XS $cm^2$ molec$^{-1}$ |
|---|---|---|
| 0 | $3.0 \times 10^{-3}$ | $14.0 \times 10^{-27}$ |
| 1 | $1.6 \times 10^{-3}$ | $5.4 \times 10^{-27}$ |
| 2 | $1.0 \times 10^{-3}$ | $4.6 \times 10^{-27}$ |

**Table 4.** Magnitude peak-to-peak (ptp) residual sizes and upper limits for water vapour absorption between 310 and 350nm at a spectral resolution of 0.7 nm for different polynomial degrees of the DOAS polynomial using the water vapour dSCD determined from *POKAZATEL* at 363 nm of $4.3 \times 10^{23}$ molec cm$^{-2}$ for the spectrum from *ANT XXVIII/1-2* shown in Figure 5. For the calculation of the upper limit we used conservatively only half of the value of the dSCD in order to account for the shorter light path at wavelengths between 310–350 nm.

dSCDs were determined using a DOAS polynomial of third order, then the dSCDs of the trace gases in the fit were fixed to these values.

The resulting upper limits for the water vapour absorption cross-section in the spectral range from 310–350 nm are thus 200–600 times smaller than the maximum cross-section values measured by Du et al. (2013) and are 14–33 times smaller than the upper limit value presented in Wilson et al. (2016).

### 4.9 Estimation of the accuracy of the $H_2O$ cross-section

The DOAS fit provides dSCDs as mentioned above, but also residual spectra $\mathbf{R}_i$. These residual spectra are the difference between the modeled and the observed OD (compare Figure 5). In order to disentangle different contributions to the residual spectra, a multi-linear regression was performed based on the retrieved dSCDs (see Lampel et al., 2015a). This allows the systematic identification of residual structures caused by each of the absorbers considered in the fit (compare Table 2). However, since potential differences between modeled and observed absorptions can be compensated by any of the other absorbers, this information cannot be used to correct a given absorption cross-section. It can yield an estimate of the accuracy of the cross-section.

For *ANT XXVIII/1-2* , the resulting spectrum from 340–380 nm which correlates with the water vapour dSCD (shown in Figure 10) has an RMS of $1.7 \times 10^{-28}$ cm$^2$ molec $^{-1}$ and a maximum peak-to-peak amplitude of $1.1 \times 10^{-27}$ cm$^2$ molec $^{-1}$. The maximum magnitude of water vapour absorption cross-section at 363 nm is $2.5 \times 10^{-27}$ cm$^2$ molec $^{-1}$ for this spectral resolution (see Figure 1). For M91, a residual structure at 344 nm is found, which could not be attributed to other absorbers and is correlated with the water vapour dSCD. As this residual structure is not observed for both datasets, we do not attribute it to water vapour absorption.

The maximum absorption of water vapour at 363 nm according to *POKAZATEL* seems to be red-shifted by 0.5 nm relative to the maximum absorption listed in BT2 (see inset in Figure 1). To test if the wavelength of the water vapour absorption is correct, a spectral shift of the water vapour absorption was allowed, i.e. the shift was determined by the Levenberg-Marquardt

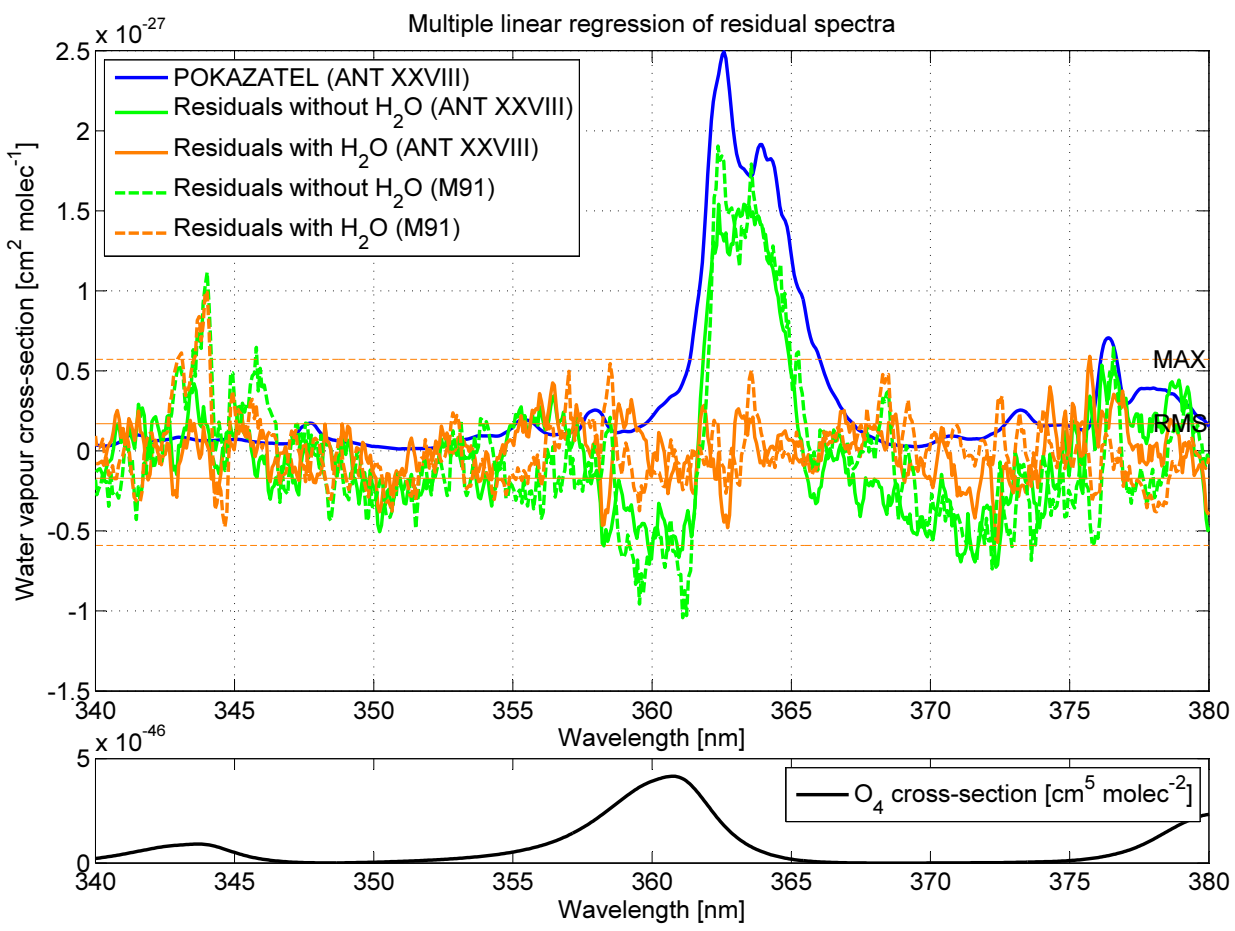

**Figure 10.** Using a multi-linear regression on the residual spectra from the campaigns *ANT XXVIII/1-2* and M91, the water vapour dSCD-correlated residual structures were obtained. Negative values can be explained by compensation of the missing water vapour absorption structures by other absorbers included in the DOAS fit. The resulting spectrum including water vapour absorption yields an estimate on the accuracy of the convolved cross-section.

algorithm of the DOAS fit. As the spectral resolution is higher, this was done for the M91 measurements. The shift of the *POKAZATEL* water vapour absorption was found to agree with observations within $0.02 \pm 0.06$ nm (corresponding to $1.5 \pm 4.6$ cm$^{-1}$) for measurements exceeding a signal to noise ratio for the water vapour dSCD of 5 for the 16 elevation sequence co-added M91 dataset. This result is in agreement with the estimate of the precision of the PES by Polyansky et al. (2016), which was able to reproduce energy levels within about 0.1 cm$^{-1}$ on average.

### 4.10 Further potential error sources

As the observed OD for water vapour absorption were small in the UV ($< 2\%$ for individual absorption lines at high spectral resolution), no saturation correction (Wenig et al., 2005) was applied during convolution of the line list for the spectral retrieval of MAX-DOAS data. The *POKAZATEL* line list does not provide line broadening parameters, therefore also the $I_0$ correction (Platt et al., 1997) was not applied. This correction would have resulted in a change of the convolved cross-section of less than $5\%$.

In the visible (452-499 nm), the saturation effect for dSCD of $6 \times 10^{23}$ molec cm$^{-2}$ amounts to less than 2% change of the obtained dSCD.

#### 4.10.1 Uncertainties of the $H_2O$ literature cross-sections

Since we compared the UV absorptions of $H_2O$ vapour to the values derived in the blue spectral region the errors in the latter spectral region - which we analyse in the following - enter into the calculation of the total uncertainty of the UV absorption cross sections of $H_2O$.

The absolute magnitude of water vapour cross-section (HITEMP) in the blue wavelength from 452-499 nm introduces an uncertainty of less than 15%: The $6\nu$ absorption band around 490 nm seems to be overestimated by $(13 \pm 3)\%$ relative to the $6\nu + \delta$ absorption band around 470 nm. This is one of the main reasons for the strongly structured fit residual in the visible fit range shown in Figure 5.

The magnitude of the $6\nu + \delta$ absorption band around 470 nm agreed with the magnitude of the $7\nu$ absorption band around 440 nm according to LP-DOAS measurements by Lampel et al. (2015b), for which in turn an agreement within 10% with independent measurements of humidity and temperature was found in the same publication.

#### 4.10.2 Uncertainties of the $O_4$ literature cross-sections

For constant atmospheric water vapour content, water vapour and $O_4$ dSCDs from MAX-DOAS observations are typically well-correlated because the bulk of the variations in the H2O-dSCD is due to variations in the path-length. Therefore it is important to disentangle potential problems of the water vapour absorption cross-section and $O_4$ absorption cross-section. The three available $O_4$ cross-sections for the spectral range below 400 nm were published by Greenblatt et al. (1990), Hermans et al. (1999) and Thalman and Volkamer (2013). The *POKAZATEL* water vapour line list shows a local maximum at 363 nm (at a spectral resolution of 0.45 nm) which is at the slope of the $O_4$ absorption peak at 360.8 nm.

Differences in differential OD from 340 nm to 390 nm between different literature $O_4$ cross-sections amount to up to $2\times10^{-3}$ for a typical dSCD of $O_4$ of $4 \times 10^{43}$ molec$^2$ cm$^{-5}$ . This is larger than the OD of water vapour in this spectral range as listed in *POKAZATEL*. A systematic error in the respective $O_4$ cross-section which could lead to false apparent water vapour absorption, which is expected to scale with the column density of $O_4$ and would thus result in a constant offset of the correlation of $H_2O$ / $O_4$ ratios shown in Figure 6. This was not observed. This also agrees with the observation that the wavelength dependence of the $O_4$ dSCDs was found to have no result on the water vapour dSCD at 363 nm. Thalman and Volkamer (2013) state an absolute accuracy of 2–4% for the their integrated $O_4$ absorption cross-section at 361 nm and 476 nm.

For strong absorbers, the AMF of the observation also depends on the magnitude of the absorption itself (Marquard et al., 2000; Pukīte et al., 2010). However, for an optical density of $O_4$ at 360.8 nm of $2.5 \times 10^{-3}$ we estimate a reduction of the effective light path by less than 1.3%, which is an OD of less than $3.5 \times 10^{-4}$ and would result in a reduction of the apparent water vapour dSCD by 10%. This effect would be smaller by a factor of 4 in tropical regions due to a smaller contribution of the $O_4$ absorption to the total optical depth. No direct correlation of the water vapour dSCDs at 363 nm with the $O_4$ dSCDs was found for the *ANT XXVIII/1-2* dataset.

| Trace gas | Wavelength nm | RMS | rel. Change of dSCD per $H_2O$ dSCD | Typ. diff. |
|---|---|---|---|---|
| $O_4$ | 340–380 | -25% | $+2.9 \times 10^{18}$ molec $cm^{-3}$ | +5% |
| HONO | 337–375 | -18% | $+1.4 \times 10^{-9}$ | +22 ppt |
| OClO | 332–370 | -20% | $+3.1 \times 10^{-11}$ | +0.5 ppt |
| $SO_2$ | 337–375 | -20% | $-2.3 \times 10^{-7}$ | -3.6 ppb |

**Table 5.** Impact on spectral retrievals estimated from DOAS evaluations with and without accounting for the water vapour absorption from *POKAZATEL* for the M91 MAX-DOAS data set (at a spectral resolution of 0.45 nm or 34$cm^{-1}$ at 362.3 nm/27601$cm^{-1}$ ). The typical difference was estimated for a water vapour dSCD of $4 \times 10^{23}$ molec $cm^{-2}$ along a 10 km long light path.

The differences between the cross-sections published by Thalman and Volkamer (2013), Greenblatt et al. (1990) and Hermans et al. (1999) did not allow to identify any systematic differences similar to the water vapour absorption, which could have pointed towards water vapour absorption during the acquisition of the cross-section data.

As seen in Table 3, it was possible to observe good correlations for water vapour absorption at 363 nm and around 477 nm for all available $O_4$ literature cross-sections. The smallest offset is observed when using the $O_4$ cross-section by Thalman and Volkamer (2013). The best correlation coefficients $R^2$ are found for Thalman and Volkamer (2013) and Hermans et al. (1999).

Absolute maximum $O_4$ absorption cross-section values differ for the three available cross-sections at 293K by less than 7% at 360 nm and less than 5% at 477 nm. This uncertainty could directly affect to the $H_2O/O_4$ ratios listed in Table 3.

### 4.11 Influence on DOAS retrievals of other trace gases

Neglecting the water vapour absorption around 363 nm increases not only the fit errors of several DOAS trace-gas retrievals, but could also introduce a systematic bias in the trace gas concentrations obtained. Trace gas species which are potentially influenced are $O_4$, HONO, OClO and $SO_2$.

The effect may vary for different data-sets, different DOAS-fit intervals and different instrumental parameters such as the respective spectral resolution. Here the impact on trace-gas retrieval is investigated based on M91 MAX-DOAS data set using the settings listed in Table 2.

#### 4.11.1 $O_4$ (340–380 nm)

For MAX-DOAS observations, the effective light path length needs to be determined to convert observed slant column densities into concentrations of the respective trace gas. The absorption of the oxygen dimer $O_4$ can be used to infer information about atmospheric light paths (e.g. Wagner et al., 2002). Atmospheric aerosol extinction profiles can be estimated by constraining the input parameters of radiative transfer models to match the observed $O_4$ column densities. For MAX-DOAS measurements this approach has been described e.g. in Wagner et al. (2004); Frieß et al. (2006). However, for some observations of scattered

sunlight, the absorption of $O_4$ had to be corrected by a correction factor in order to explain the measured column densities as reported by (e.g. Wagner et al., 2009; Clémer et al., 2010; Irie et al., 2015). Clémer et al. (2010) estimated a correction factor value of $1.2-1.5$ for modelled differential slant column densities (dSCD) values to match observed dSCDs. The reason for this correction factor is so far unknown. However, for direct-sun DOAS measurements and measurements in the tropopause (Spinei et al., 2015) showed that a correction factor is not necessary to explain the measurements. Recently a possible explanation for a part of these previous observations was provided by Ortega et al. (2016): elevated aerosol layers in heights above 2 km which affected the apparent $O_4$ dSCDs but could not be resolved from ground-based MAX-DOAS measurements due to their limited information content for aerosol extinction in these altitudes. Another reason for this correction factor could be an unaccounted tropospheric absorber, such as e.g. water vapour absorption.

To estimate the effect of water vapour absorption, the same evaluation for $O_4$ according to Table 2 was performed once with and once without the *POKAZATEL* water vapour absorption cross-section. An increase in $O_4$ dSCD is observed when including the *POKAZATEL* water vapour absorption cross-section in the DOAS evaluation.

Using the correction factor of 2.63 determined in Figure 4.2, including the water vapour absorptions leads to an increase in $O_4$ dSCD per $H_2O$ dSCD of $+(2.9 \pm 0.3) \times 10^{18}$ molec cm$^{-3}$, independent of the settings whether a shift and/or squeeze is allowed for the literature absorption cross-sections.

For a typical $H_2O$ dSCD of $4 \times 10^{23}$ molec cm$^{-2}$ in summer at mid-latitudes and a $O_4$ dSCD of $2.5 \times 10^{43}$ molec$^2$ cm$^{-5}$ (10 km light path length) including the water vapour absorption leads to an absolute increase of $O_4$ dSCD of $1.2 \times 10^{42}$ molec$^2$ cm$^{-5}$ , which corresponds to a change of $+5.0\%$.

Thus the water vapour absorption at 363 nm cannot explain the correction factor for $O_4$ dSCDs introduced in various publications (see subsection 1.2), it even increases the factor by +5.0% for measurements during summer in mid-latitudes.

#### 4.11.2 HONO (337–375 nm)

Nitrous acid (HONO) is a key species in the atmospheric chemistry of urban air-masses (e.g. Perner and Platt, 1979), because its photolysis leads to the production of OH radicals, the 'detergent' of the atmosphere. Due to its high reactivity and fast daytime photolysis, HONO concentrations are low, in particular during daylight hours (Wong et al., 2012), and thus their measurements are difficult, but can be performed e.g. by absorption spectroscopy. If all relevant absorbers are accounted for, spectroscopic measurements have the advantage of being less affected by interferences, which were observed for wet chemical methods, such as e.g. LOPAP (e.g. Kleffmann and Wiesen, 2008). Therefore it is important to account for all possible absorbing trace gas species in the respective wavelength range, e.g. 337–375 nm (Hendrick et al., 2014), in order to further reduce the detection limit and eliminate potential biases.

Adapting the wavelength range from (Hendrick et al., 2014, and using the settings listed in Table 2), neglecting the water vapour absorption in the HONO fit has led to an decrease of HONO dSCDs. The decrease is clearly correlated to the water vapour dSCD and amounts per corrected $H_2O$ dSCD to $1.4 \times 10^{-9}$. This corresponds for a $H_2O$ dSCD of $4 \times 10^{23}$ molec cm$^{-2}$ to a negative bias of HONO dSCDs by $5.6 \times 10^{14}$ molec cm$^{-2}$ , which corresponds to a HONO surface volume mixing ratio of 22 ppt along a light path of 10 km.

The RMS decreases for this water vapour dSCD by $0.4 \times 10^{-4}$ at a typical RMS of $2.2 \times 10^{-4}$, which is a decrease of 18%. This decrease of dSCDs explains negative HONO dSCDs around noon during M91, when not considering water vapour absorption.

At an elevation angle of 3° we obtain a distribution of dSCDs around $(-3.9 \pm 2.4) \times 10^{14}$ molec cm$^{-2}$ without including water vapour absorption. Including the water vapour absorption, the HONO dSCDs are distributed around $(1.0 \pm 2.3) \times 10^{14}$ molec cm$^{-2}$ . During the cruise significant positive HONO dSCDs were observed close to $NO_2$ plumes from cities (HONO dSCDs of up to $2 \times 10^{15}$ molec cm$^{-2}$ at low telescope elevation angles), when the cruise track was close to the Peruvian coast. Therefore a slightly positive average HONO dSCDs can be expected.

#### 4.11.3 OClO (332–370 nm)

Stratospheric OClO has been observed in polar regions (e.g. Solomon et al., 1987; Kühl et al., 2008; Oetjen et al., 2011). Recently, OClO has also been observed in volcanic plumes (Bobrowski et al., 2007; Theys et al., 2014; Donovan et al., 2014; General et al., 2015; Gliß et al., 2015). All of these measurements were limited on one side of the retrieval interval close to 360 nm, potentially indicating unaccounted absorptions or erroneous $O_4$ cross-sections. Saiz-Lopez and von Glasow (2012) and references therein suggested that so far tropospheric OClO outside volcanic plumes has been observed only in polar regions with small absolute tropospheric water vapour content.

The 363 nm water vapour absorption band is located between two absorption bands of OClO and thus neglecting the water vapour absorption leads to an underestimation of OClO dSCDs and systematic residual structures.

Even when including water vapour absorption according to *POKAZATEL*, OClO was not positively identified during M91 (332–370 nm) above a $2\sigma$ detection limit of $1.6 \times 10^{13}$ molec cm$^{-2}$ at an elevation angle of 3°, the dSCDs showed a distribution around $(-0.9 \pm 8.0) \times 10^{12}$ molec cm$^{-2}$ . Without correction for water vapour absorption the dSCDs showed a distribution around $(-6.3 \pm 8.9) \times 10^{12}$ molec cm$^{-2}$ .

Corrected by the scaling factor of 2.63 from Figure 4.2, the increase in OClO dSCD per $H_2O$ dSCD amounts to $3.08 \times 10^{-11}$. The difference in OClO is clearly correlated with the $H_2O$ dSCD with $R^2 = 0.9$. This corresponds for a $H_2O$ dSCD of $4 \times 10^{23}$ molec cm$^{-2}$ to an increase of OClO dSCD by $1.2 \times 10^{13}$ molec cm$^{-2}$ , which corresponds to a OClO surface volume mixing ratio of 0.5 ppt along a light path of 10 km.

#### 4.11.4 Impact on the retrieval of other absorbers

In the spectral region below 360 nm, concentrations of HCHO and BrO can be retrieved. For HCHO systematic problems were discussed in Pinardi et al. (2013) and pointed towards uncertainties of the available $O_4$ cross-sections. The absorptions listed within this fit range (336.5-359 nm) in BT2 are of similar magnitude as BrO concentrations for the lower troposphere as reported by (Richter et al., 2002; Volkamer et al., 2015). *POKAZATEL* also lists lines here. So far, the absorption at 335 nm could not be unambiguously identified in measurements but can potentially have an impact on the spectral retrievals of tropospheric BrO and HCHO (see subsection 4.6).

For very high column densities of $SO_2$, alternative DOAS evaluation wavelength intervals above 340 nm can be used in order to minimize saturation effects due to large optical depths (Bobrowski et al., 2010; Hörmann et al., 2013). If such spectral evaluation schemes are applied to ground-based MAX-DOAS measurements using also low telescope elevation angles for locations with high absolute water vapour concentrations, water vapour absorption might need to be considered also in the spectral evaluation of $SO_2$. We estimated the impact using the HONO (337–375 nm) fit settings with the additional $SO_2$ absorption cross-section from Vandaele et al. (2009) in Table 5. The overall change in dSCD was of the same magnitude as the fit error itself.

### 4.12 MAX-DOAS: Relative water vapour absorption band strengths in the blue spectral range

The consistency of the *POKAZATEL* line list with other line lists and measured absorption was checked in analogy to Lampel et al. (2015b) in the blue spectral range for MAX-DOAS observations. The relative absorption strength relative to the much stronger absorption around 442 nm was determined for the *POKAZATEL* water vapour line list. The different wavelength intervals are listed in Table 6. The same MAX-DOAS data set (M91) and the same settings as described in Lampel et al. (2015b) were applied. The magnitude of the absorptions W0 and W1 are underestimated compared to MAX-DOAS observations, leading to the observation of water vapour dSCDs, which are 26%(W0) and 71%(W1) larger than the dSCDs observed simultaneously for the stronger absorption W3. The results are shown in Table 7 and Table 8.

Overall, *POKAZATEL* predicts the integrated absorption cross-sections in the blue spectral until 480 nm range better than previous versions of HITRAN and BT2, as seen from Table 6 and summarized in Figure 11. It was however not used as a reference cross-section in the blue wavelength range, as HITEMP (and HITRAN2012) reproduced the observed water vapour absorptions in the blue fit interval (452–499 nm) significantly better. These differences which are also seen from Figure 1 will require further investigation, as they do not only involve a difference of the overall absorption strength of both bands near 470 and 490 nm, but also differences in the shape of the absorption bands were observed between HITEMP and *POKAZATEL* (see also subsubsection 4.10.1).

## 5 Conclusions

The water vapour absorption structure predicted from calculations for wavelengths around 363 nm by Polyansky et al. (2016) was found for the first time in two different MAX-DOAS measurement data-sets of tropospheric air-masses with optical depths of up to $2 \times 10^{-3}$ at a spectral resolution of 0.45–0.7 nm. Additionally it was observed for the first time in LP-DOAS observations. Until now, to our knowledge these absorptions were neither experimentally verified nor considered in the spectral analysis of DOAS observations.

Comparing the strenghts of the UV absorption lines of water vapour to the water vapour absorptions listed in HITEMP between 452 and 499 nm showed that the absorptions are indeed caused by water vapour and that the cross-section calculated from the data provided by Polyansky et al. (2016) underestimates the measured absorption by a factor of $2.6 \pm 0.5$. For MAX-DOAS, the different light path lengths in the two different wavelength windows were corrected by normalization with

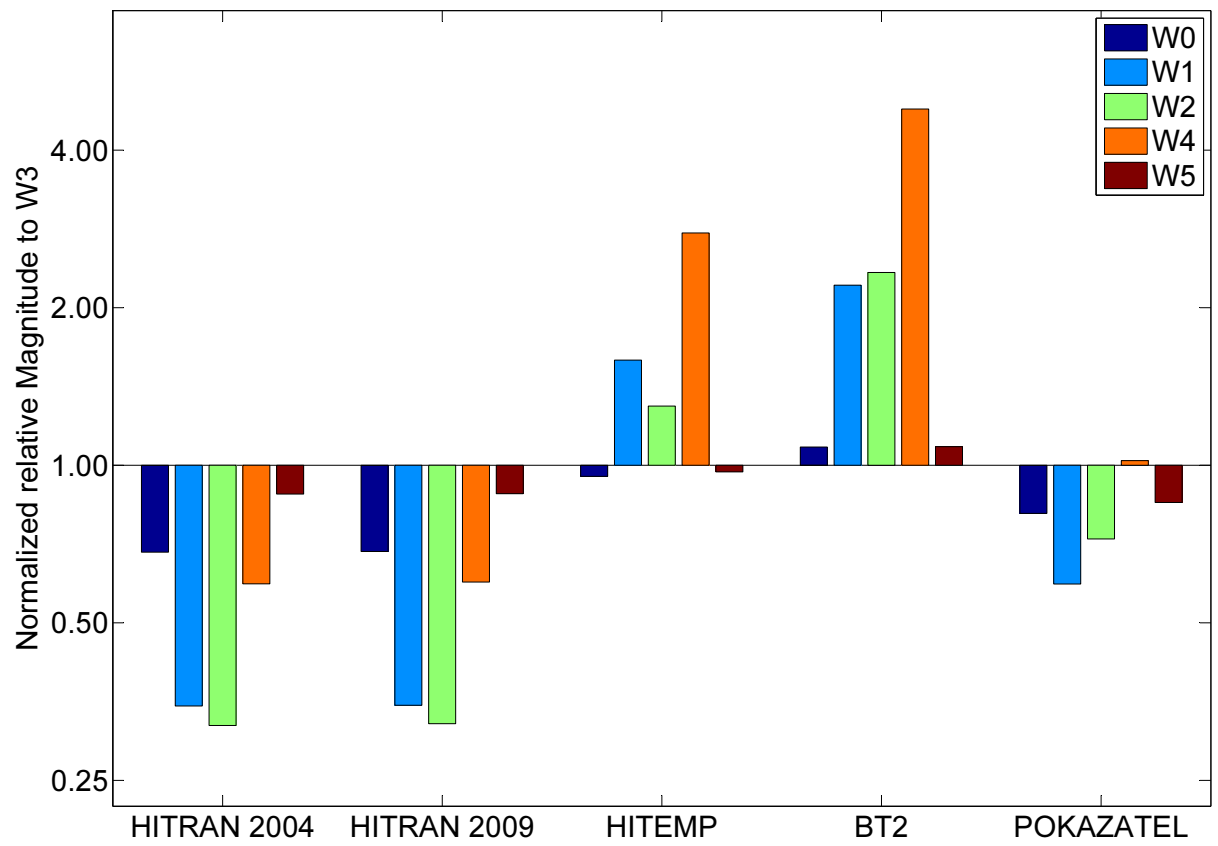

**Figure 11.** Comparison of different available water vapour cross-section data in the blue spectral range using the different band listed inTable 6. W3 was used as a respective reference in all cases and is therefore by definition unity. All magnitudes were normalized with respect to the rescaled HITEMP absorption cross-section from Lampel et al. (2015b) to obtain relative magnitudes of each of the absorption bands W0,W1,W2,W4 and W5. A value of '1' identifies good agreement with the relative magnitude of the absorption bands' sizes according to MAX-DOAS and LP-DOAS measurements presented in Lampel et al. (2015b).

the respective $O_4$ dSCD in the same wavelength interval. The water vapour absorption feature at 363 nm in MAX-DOAS measurements was identified and shown to be independent of the chosen literature value of the $O_4$ absorption cross section, i.e. it was found to be at a similar magnitude for all three available $O_4$ literature cross-sections. It was also independent of the temperature-induced broadening of the $O_4$ cross-section.

In contrast, a slight spectral shift of the $O_4$ reference spectrum could have compensated in previous evaluations (not including the 363 nm $H_2O$ absorption) for the water vapour absorption, which is located on a slope of the $O_4$ absorption (subsection 4.5). This apparent shift might have lead to wavelength calibration corrections of $O_4$ literature cross-sections in previous publications for individual campaigns with relatively constant $H_2O/O_4$ dSCD ratios.

Other predicted water vapour absorption features at 335 nm could not be unambiguously identified in the measurements as they did not exceed the respective detection limits. The absorption structure at 377 nm was slightly above the detection limit and was found to correlate with the water vapour absorption at 363 nm.

The identified water vapour absorption at 363 nm can have a significant impact on the retrieval of trace-gases, which absorb in the same wavelength range, namely $O_4$, HONO, OClO and $SO_2$. For measurement locations with high absolute water vapour concentrations, consideration of the water vapor absorption at 363 nm, if included in the spectral analysis of MAX-DOAS measurements, will lead to a reduction of measurement errors and will thus lower the overall limit of detection. We showed that neglecting this absorption introduce systematic biases in their spectral analysis:

| Dominating Polyad | | $8\nu$ | $7\nu+\delta$ | | $7\nu$ | | $6\nu+\delta$ |
|---|---|---|---|---|---|---|---|
| Name | | W0 | W1 | W2 | **W3** | W4 | W5 |
| Start of interval | [nm] | 394.0 | 410.0 | 423.5 | **434.0** | 451.5 | 461.5 |
| End of interval | [nm] | 410.0 | 423.5 | 434.0 | **451.5** | 461.5 | 480.0 |
| Source of cross-section data | [$10^{-27}$ nm cm$^2$] | integrated cross-section | | | | | |
| HITRAN 2000 | | 0.00 | 0.00 | 0.00 | **69.02** | 0.00 | 31.03 |
| HITRAN 2004 | | 13.62 | 3.11 | 0.89 | **96.75** | 0.87 | 42.25 |
| HITRAN 2009 | Rothman et al. (2009) | 13.71 | 3.13 | 0.90 | **97.07** | 0.88 | 42.46 |
| HITEMP | Rothman et al. (2010) | 21.01 | 15.73 | 4.01 | **106.90** | 4.50 | 51.44 |
| BT2 | Barber et al. (2006) | 26.05 | 23.84 | 7.86 | **116.50** | 8.46 | 62.67 |
| HITEMP rescaled | Lampel et al. (2015b) | 22.06 | 9.91 | 3.09 | **106.90** | 1.62 | 52.98 |
| *POKAZATEL* | Polyansky et al. (2016) | 15.98 | 5.26 | 2.00 | **95.7** | 1.48 | 40.26 |

**Table 6.** Integrated absorption in [$10^{-27}$ nm cm$^2$] over each of the wavelength intervals W0-W5 for different sources of cross-section data. Not only for the largest absorption structure W3 variations between the different compilations are seen, but especially relative integrated absorption values vary. The last row shows the maximum optical density for a water vapour column density (CD) $S = 4\times10^{23}$ molec cm$^{-2}$ within each wavelength interval at a spectral resolution of 0.5 nm for HITEMP. The upper part of this table is adapted from Lampel et al. (2015b). This data is visualized in Figure 11.

During M91, for $O_4$ dSCDs an increase of about 5% was observed when including the additional absorption in the DOAS analysis. Thus, the water vapour absorption cannot explain the much larger correction factor for $O_4$ dSCDs introduced in various publications (it rather increases the observed discrepancies).

For HONO the water vapour absorption explains negative HONO dSCDs of several $10^{14}$ molec cm$^{-2}$ for mid-latitude absolute water vapour volume mixing ratios. Negative HONO dSCD at low elevation angles were often observed around noon during the SOPRAN M91 campaign in the Peruvian upwelling when not considering water vapour absorption. In the same way negative OClO dSCDs in MAX-DOAS observations at low elevation angles of around $-1\times10^{13}$ molec cm$^{-2}$ during M91 could also be linked to water vapour absorption at 363 nm.

Future DOAS evaluations encompassing the spectral range around 363 nm will require to include this water vapour absorption features, if they aim at residual spectra with an RMS of less than $4\times10^{-4}$ or try to fit absorbers with measurement errors corresponding to optical densities of less than $1\times10^{-3}$ in mid-latitude to tropical regions.

The predictions of *POKAZATEL* do not yield complete agreement with the observed absorption features. While, as discussed above, this line list should give very accurate line positions, the situation regarding absorption intensities is more problematic. This is indeed observed in the measurements presented here, as the position of the absorption is found to be accurate (shift of $0.02 \pm 0.06$ nm, or $1.5 \pm 4.6$ cm$^{-1}$), while the magnitude of the observed absorption bands differs relative to each other. This was before also observed in the blue spectral range by Lampel et al. (2015b). While the *ab initio* dipole moment calculations of Lodi et al. (2011) cover an appropriate range of geometries and are expected to be accurate, using them to construct a reliable DMS is not straightforward. A number of studies (Schwenke and Partridge, 2000; Lodi et al., 2008; Tennyson, 2014) have shown that it is difficult to produce analytic fits which correctly reproduce the intensity of weak transitions. Here we are dealing with very weak water absorptions on the margins of detectability. For this reason we performed some test calculations using the *POKAZATEL* methodology but utilizing the CVR DMS of Lodi et al. (2008). The results shown in subsection 4.3 indicate that this DMS (Lodi et al., 2008) could explain the systematic underestimation of the magnitude of water vapour absorption, but probably do not predict the spectral shape of the absorption peak as accurately as *POKAZATEL*. Further work is required on the precise representation of the *ab initio* DMS to try to resolve these problems. Studies should also be performed to obtain a more reliable representation of the water dipole moment for the purpose of predicting absorption intensities in the near UV. Laboratory studies on this problem would also be very helpful.

The values for the absorption cross section of water vapour in the UV range reported by Du et al. (2013) cannot be confirmed. We derived upper limits, which are at least two orders of magnitude smaller in the spectral range from 310–370 nm.

## Appendix A: Relative absorption strengths

| Name | | W0 | W1 | W2 | **W3** | W4 |
|---|---|---|---|---|---|---|
| Start of interval | [nm] | 394.0 | 410.0 | 423.5 | **434.0** | 451.5 |
| End of interval | [nm] | 410.0 | 423.5 | 434.0 | **451.5** | 461.5 |
| *POKAZATEL* | | 1.2605(6) | 1.7052(13) | [0.8135(41)] | **1** | [2.1270(81)] |

**Table 7.** Measured relative absorption band strengths for the different cross-sections with respect to the absorption at W3, the $7\nu$ polyad, column in bold face. Errors obtained from the linear regression are shown for the last digits in brackets. The relative DOAS fit errors are listed in Table 8. Results with typical DOAS fit errors of more than 25% of the measured values were put in square brackets. MAX-DOAS values are corrected by the results of radiative transfer modelling Lampel et al. (2015b).

*Acknowledgements.* We thank the captain, officers and crew of RV Polarstern for support during cruise ANT XXVIII. Especially for the support by J. Rogenhagen/FIELAX/AWI and technicians on board.

| [%] | | W0 | W1 | W2 | **W3** | W4 |
|---|---|---|---|---|---|---|
| Start of interval | [nm] | 394.0 | 410.0 | 423.5 | **434.0** | 451.5 |
| End of interval | [nm] | 410.0 | 423.5 | 434.0 | **451.5** | 461.5 |
| *POKAZATEL* | MAX-DOAS | 4 | 6 | 40 | **0.8** | 29 |

**Table 8.** Typical relative DOAS fit errors in fitting windows W0-W5 at a water vapour dSCD in W3 of $4 \times 10^{23}$ $\mathrm{molec\,cm^{-2}}$ for an individual spectrum integrated over 60 s. Values are given in % and are corrected by the relative magnitudes given in Table 7.

We thank the captain, officers and crew of RV Meteor for support during cruise M91.

We thank the German Science foundation DFG within the core program METEOR/MERIAN. We thank the German ministry of education and research (BMBF) for supporting this work within the SOPRAN (Surface Ocean Processes in the Anthropocene) project (Förderkennzahl: FKZ 03F0662F) which is embedded in SOLAS.

We thank the UK Natural Environmental Research Council and ERC, through Advanced Investigator Award 267219, for partial support of this project.

We thank the Russian Fund for Fundamental Studies grant No. 15-02-07473 A.

We thank Holger Sihler for helpful discussions during the preparation of the manuscript and Cornelia Mies and Rüdiger Sörensen for technical support and data processing.

We thank GEOMAR for logistical support.

We thank the authorities of Peru for the permission to work in their territorial waters.

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
