# Peer review of "Detection of water vapour absorption around 363 nm in measured atmospheric absorption spectra and its effect on DOAS evaluations"

_Atmospheric Chemistry and Physics, 2016_

## Referee Comment (RC1) · Anonymous Referee #1 · 26 Jul 2016

Lampel et al. report new experimental observations for water absorption bands below 390 nm and consider how this water absorption influences the retrieval of other atmospheric species in a spectral fit. Water absorption in the near-UV region has received significant attention in the last few years, notably with a report of significant water vapour absorption below 360 nm, contrary to theoretical predictions of decreasing water absorption strength at shorter wavelengths. That report has since been called into question, leaving no clear experimental evidence for water absorption at such short wavelengths, despite theoretical predications of several very weak absorption bands. Using very long optical pathlengths through the atmosphere in both LP-DOAS and MAX-DOAS measurements, Lampel et al. convincing verify a water absorption band

around 363 nm, both through the strong correlation between this band and a much stronger, well attested water absorption at longer wavelengths, and through the excellent match with the expected band structure and position of the latest theoretical line list (POKAZATEL, 2016). Similar evidence was presented for another water absorption band at 376 nm, but other water bands (including a predicted band around 335 nm) could not be confirmed. The magnitude of predicted water absorption in the 363 nm and 376 nm bands was too low by a factor of 2 – 3. The focus of the paper then turns to the effect of the 363 nm water absorption band on the spectral analysis and quantification of other molecular species in the near-UV. These include O4, HONO, OClO, and SO2. The impact on water absorption on these retrievals is not large, but nonetheless significant enough to warrant inclusion in future retrievals for these long open path measurements.

This is a comprehensive & multifaceted study of water absorption in this spectral region and I have no particular concerns about the analysis and conclusions of the paper. The water absorption is confirmed in three distinct data sets with large differences in the water slant column densities. This approach is necessary given the small magnitude of water absorption in the experimental spectra. The authors take considerable pains to rule out other confounding factors in the spectral analysis, which include wavelengths shifts in the O4 band, differences between experimental and theoretical spectra. The effects of different atmospheric structure on radiative transfer are also simulated. These experimental and analytical results are internally consistent within the uncertainties of the measurement.

What may be valuable for future work on radiative transfer and theoretical studies on the water molecule absorption, is some discussion in the paper of whether it is possible to obtain more detailed experimental measurements of water absorption lines in the near-UV. In particular, would such an analysis be possible and more sensitive with a higher resolution system?

Moreover, much of the initial impetus for measuring the water absorption spectrum was

concerned with radiative transfer in the atmosphere. What implications does this paper bring to bear on that question?

Possibly owing to the variety of topics explored, this is not an easy paper to read. Nevertheless, the standard of editing falls short of ACP standards and should be addressed. Some obvious errors are listed in the technical corrections, and I encourage the authors to review the text carefully again.

Technical corrections:

1. Reported physical properties should have a space between the value and the units. This is not consistently adhered to in the manuscript.

2. Reference needed: p.4, l.22 after "unaccounted tropospheric absorber"

3. P.7, l.2-6: It is unclear whether the absorption cross section refers to total cross section, or to the differential cross section. The symbols used are those conventionally used for total absorption cross section. See e.g., Platt, Phys. Chem. Chem. Phys., 1999, 1, 5409-5415, for the usual description.

4. P23, l18: Do the authors have an explanation for the residual feature observed in one dataset?

5. The quantities described in Table 5 are not sufficiently clear to this reviewer, and the columns should be more precisely defined than "impact". If, as I presume, what is meant is (e.g.) the difference between RMS (water absorption included) – RMS (no water absorption), then this should be stated. Likewise for the other properties.

The following parts of the document should be edited:

6. Abstract:

a. "visible spectrum at a decreasing" . . . "visible spectrum with decreasing"

b. "until its dissociation limit" . . . "up to its dissociation limit"

7. Page 1:

a. 15: "vapour. it plays a key role for the"..." vapour. It plays a key role in the"

b. 16-17: "Earth...absorption" ...unclear.

c. 19: "also required assessing"..." also required for assessing"

8. Page 4:19: "and SO2 , potentially even HCHO and BrO". Unclear

9. Page 5: 20: "Bremerhaven/....employed". Unclear

10. Page 7:

a. 3: "and narrow-band"..." and a narrow-band"

b. 15: ""measurements is, that ... "measurements is that"

11. Page 8:

a. 8: "Longpath(LP)-DOAS" and "(here a a LASER-driven"

b. 15: Unclear.

12. Page 12:19: "Due to need to"

13. Page 14: 2: "selected such according". Unclear

14. Page 21: 27: "water cross-section of O3 ,"..." absorption cross-section of O3 ,"

15. Page 29: 29: "strenghts"

---

## Referee Comment (RC2) · Anonymous Referee #3 · 21 Oct 2016

The paper discusses important new information about water vapour absorption in the UV spectral region and its effects on DOAS retrievals. With this it presents important results which are suitable for publishing in ACP. However, the presentation is at many points confusing and I suggest publication only after major revisions. Due to the lengths of this paper of 40 pages the content is hard to follow and this isn't helped by the fact that section titles don't always fit the content (e.g. section 4.9 is about the accuracy of the wavelengths axis and this should be spelled out in the title; section 1.3 lists science questions and not an outline). Another problem is that the authors clearly have lost track themselves, e.g.: there is no proper introduction about the differences of HITRAN 2009, 2012, HITEMP, or BT2 in the beginning of the manuscript, but there are bits and

pieces of information later on in the text; the lower panels in Figure 1 are not referred to in the text at all; the text refers to a figure 4.2 which I believe is actually Figure 6; on the other hand, there is no reference in the text to the right panel in Figure 4; the spectral resolution of the instruments is stated 3 times in the manuscript, but some important information is only listed in captions, e.g. how the upper limit is calculated for Table 4; there are 2 different symbols used for absorber concentrations in the equations. More details/corrections below. However, I would like to encourage the authors to give this manuscript a thorough read themselves and to restructure some of its content, especially double-checking if the information provided in the figures/tables and their captions is actually used and sufficiently described in the manuscript.

Specific comments:

p.1, l.5, 11, 13: 363 nm or 362.3 nm. I understand that the authors refer to the peak of the absorption and the feature in general. But using two different numbers without further explanation in the abstract is confusing.

p.1, l.8: Add: 'For MAX-DOAS measurements, we observed...'

p.1, l.8: 'It correlates...' refers to something like 2 months of data. That should be made clear at this point.

p.1, l.10: Add: '...line intensities at 362.3 nm are underestimated by...'

p1., l.12: 'spectral retrievals'

p.1, l.15: 'It'

Figure 1, top panel: The y-axis on the inset plot seems to have a different extent, especially at the lower end. This makes it appear as if the POKAZATEL has more lines in that inset than in the large plot.

p.3, l.6 and following: Why not refer to Figure 1 at this point already? The intro would read easier if you structured it like this: In Lampel et al. (2015b) you already suspected

additional water vapour lines. But those could not be found in the line list available back then. Describe the available line list until then. Then you introduce the new POKAZATEL list and that this new information will be investigated with additional field measurements. Splitting the intro into subsections actually interrupts the flow of the argument.

p.3, l.16-17: Add that those were lab measurements.

p.3, l.20: 'structures in the spectral range'

p3., l.24: 'individual line cut-off': what is that?

p.3, l.28-29: No new paragraph needed here

p.3, l.34: '(in principle)': Any explanation what this refers to or remove?

p.4, l.15: I would remove 'potential' here

p.4, l.18,21: O4 was already used without being introduced as were the other species.

p.4, l.22: Please add reference for 'unaccounted tropospheric absorber'.

p.4, l.23-26: This statement at this point is difficult to understand for a person not very familiar with MAX-DOAS measurements and the corresponding radiative transfer. Either remove or give more explanation. See also below.

p.5, l.2: formatting: brackets should be within the sentence.

Section 2: Maybe add here that LP-DOAS is active and MAX-DOAS a passive technique

p.5, l.12: remove space after 15.

p.5, l.15: Full stop after 0.45nm.

p.5, l.16: The latitudinal extent has nothing to do with variations of the water vapour mixing ration. Also, Figure 2 is misleading since the satellite in the background is from

a different time than the M91 cruise.

p.5, l.17-19: S.a.

p.5, l.20: 'therefore': s.a. water vapour changes because it changes and not because the measurements are at a different latitude.

p.5, l.22-24: Maybe make that 2-3 sentences. Your point that O4 and H2O absorb in similar region in the UV as well as in the visible doesn't fully come across. You could also refer to Figure 1 here.

p.5, l.3: lambda is not introduced

Figure 2: It's not clear how you get from a slant column ratio to a vertical column of one of the species. Also these results are not discussed in the manuscript.

p.7, l.1: I0(lambda) is introduced a second time

p.7, l.2: OD is not introduced yet

p.7, l.3: why only 'partly'?

Eq (1), (2): please use same symbol for concentration

Eq (1): add a bracket to indicate the summation; the polynomial p(lambda) is a different one than the one introduced in the line 3 above for the measurements and the cross section here should be a 'differential' cross section

p.7, l.9: See above; maybe add somewhere before already that MAX-DOAS measures scattered sunlight and LP-DOAS is an active technique.

p.7, l.12: 'spectral width'

p.7, l.14-15: The sentence about the residual is confusing at this point. Maybe remove?

p.8, l.4-6: The total light path is from the institute to the train station and back to the institute?

p.8, l.6: The spectral resolution is redundant information here. Was mentioned before.

Section 3.1: I suggest swapping the first 2 paragraphs.

p.8, l.11: Not the measurement sequence but the correction with the background spectra ensures the independence.

p.8, l.17: 'high-pass filtered literature cross-sections': aha! That should be mentioned before.

p.8, l.27: s.a., symbol for concentration.

p.8, l.31: A Fraunhofer spectrum always refers to the extra-terrestrial spectrum of the sun (or another star).

Table 2: What is this 'Add. Polynomial degree'?

p.10, l.1: Fraunhofer: s.a. and another time below as well.

p.10, l.2: full stop after bracket

p.10, l.3: Remove the last 'the' of the line

p.10, l.6: ANT XXVIII/1-2 or ANT XXVIII? Please unify in manuscript.

p.10, l.16: spectral resolution is redundant information

p.10, l.20: Why Figure 5 before Figures 3 & 4?

p.10, l.21: $40°$ telescope angle.

p.10, l.21: 'Spectra recorded at. . .': why not remove the sentence in l.1-2, p.10 then?

p.10, l.25: Are those the dSCD measurement errors? Please clarify. Also state that this disregards possible systematic errors.

p.10, l.26: This section is about the DOAS spectral fitting. So a reference to a 'linear fit' is confusing here. Please add more explanation or move/remove this sentence.

p.11, l.25: OD; s.a.

p.11, l.28-30: Please elaborate or state reference.

p.12, l.3: 'when co-adding spectra from more than'

p.12, l.12: Remove paragraph break.

p.12, l.16: Is the stated time period different from the one in Table 1?

p.12, l.20: Are you really losing 15 min for each hour? Please clarify.

p.12, l.25: The data in Figure 3 does not support this statement. Also, this is the only time Figure 3 is mentioned in the text of the manuscript (besides in the caption for Figure 2, but Figure 2 is barely mentioned either. Those would be candidates for removing in order to shorten the manuscript.).

p.12, l.29: 20% is not low humidity?

p.13, l.2: Maybe remove the uncertainty estimate at this point since it has just been stated in the line above and the actual interesting number is 0.7 and not 0.05.

Figure 4, caption: 2.31? The text states 2.4. The right panel of the figure is not mentioned in the text at all.

p.14, l.1-3: Please rephrase.

p.14, l.10: The figure states R2 = 0.74 for both cases. Please clarify!

Figure 6, caption: There is only 1 error bar and that is attached to the linear fit. Does it refer to the error bars of the measurements though? Please clarify! For the green box in the top right panel, how are the measurement uncertainties combined? The figures say O4 at 476nm and not 477 as in the caption.

p.17, l.5-7: I don't understand this paragraph. Worse in comparison to what? Did you state the number of the combined correlation somewhere? 0.91 is a pretty good correlation.

p.17, l.8-11: This information should already be stated on p.14,l.9-10. Also maybe mention somewhere that this is the reason for the different numbers for n in Table 3.

p.17, l.14: 'see also Table 3'

p.17, l.16: 'latitudinal' s.a.

p.18, l.16: 'includes more measurements': see comment above

p.18, l.31: could not be identified for either of the two line lists or cruises? Please clarify!

Figure 7, caption: add space after DMS

p.19, l.12: 'an RMS'

p.19, l.15: 'than in either the BT2 or the HITEMP'

p.19, l.15-18: This information should have been in the intro .

Figure 8: gridlines would be helpful in this figure.

p.22, l.18: '3° elevation angle'

p.22, l.16: 'polynomials with degrees 0-2 were applied in order to test...'

Table 4: Last sentence of the caption seems to be quite important, however, is not explained in the text.

Section 4.9, title: Please add that you investigate the accuracy of the wavelength calibration here

p.23, l.7: why is there an R introduced for the residual spectra? It's not used anywhere else.

p.24, l.5: Within 0.1 cm-1 in comparison to what?

p.25, l.1: Section title is misleading. This section only refers to the visible range.

p.25, l.5: '... magnitude of the water vapour... blue wavelength range...'

p.25, l.5-8: Where does this information come from?

P.25, l.14: formatting issue H2O-dSCD

p.25, l.21-23: Please split sentence

p.25, l.27-29: Please split sentence

p.25, l.30-31: 'No direct correleation...' I don't understand this. Please elaborate.

p.26, l.1-3: Maybe use the term water vapour contamination here.

p.27, l.2: formatting issues for references

p.27, l.4: formatting issues for reference

p.27, l.20: section 1.2 does not mention the correction factor

p.27, l.30: formatting issues for reference and brackets

p.28, l.5-8: Why didn't you perform this analysis separately then for cases with and without HONO?

p.29, l.1: 'alternative' to what?

p.29, l.7: remove 'itself'

p.29, l.16: rephrase sentence

p.29, l.22: Maybe join these two sections?

Figure 11, caption: 'different bands listed in Table6'; use unity instead of '1'.

Table 6, caption: 'relative integrated absorption values': relative to what? Please elaborate; The second to last, not last row shows the scaled HITEMP data.

p.31, l.10: Who are 'they'?

Table 7, caption: formatting issues with reference

Appendix: Why are table 7 and 8 in an appendix? Then the text discussing those should also be moved to the appendix.

Table 8: I'm not sure what is done here. What are the 'relative DOAS fit errors'? There is no Window 5 in here.

More general comments:

* The abstract doesn't list anything about the O4 studies or the failed identification of other water vapour lines in the UV

* References to dissertations cannot be accessed if no link is provided.

———————————————————

---

## Author Comment (AC1) · 27 Nov 2016

Dear editor, we like to thank the reviewers for their helpful comments, which we answer as detailled below.

The first review listed a number of minor revisions, while the second review suggested major revisions consisting of a larger number of revisions. We answered all of the listed points and answered some of the suggestions given in the respective introduction of the review. Finally we corrected the edited manuscript again.

(Numbers of equations, figures, lines and pages refer to the discussion manuscript, if not mentioned otherwise. Authors' reponses are written in **bold face**, the referees' text is shown in normal face.)

**1 Referee #1**

Lampel et al. report new experimental observations for water absorption bands below 390 nm and consider how this water absorption influences the retrieval of other atmospheric species in a spectral fit. Water absorption in the near-UV region has received significant attention in the last few years, notably with a report of significant water vapour absorption below 360 nm, contrary to theoretical predictions of decreasing water absorption strength at shorter wavelengths. That report has since been called into question, leaving no clear experimental evidence for water absorption at such short wavelengths, despite theoretical predications of several very weak absorption bands. Using very long optical pathlengths through the atmosphere in both LP-DOAS and MAX-DOAS measurements, Lampel et al. convincing verify a water absorption band around 363 nm, both through the strong correlation between this band and a much stronger, well attested water absorption at longer wavelengths, and through the excellent match with the expected band structure and position of the latest theoretical line list (POKAZATEL, 2016). Similar evidence was presented for another water absorption band at 376 nm, but other water bands (including a predicted band around 335 nm) could not be confirmed. The magnitude of predicted water absorption in the 363 nm and 376 nm bands was too low by a factor of 2 3. The focus of the paper then turns to the effect of the 363 nm water absorption band on the spectral analysis and quantification of other molecular species in the near-UV. These include O4, HONO, OCIO, and SO2. The impact on water absorption on these retrievals is not large, but nonetheless significant enough to warrant inclusion in future retrievals for these long open path measurements. This is a comprehensive & multifaceted study of water absorption in this spectral region and I have no particular concerns about the analysis and conclusions of the paper. The water absorption is confirmed in three distinct data sets with large differences in the water slant column densities. This approach is necessary given the small magnitude of water absorption in the experimental spectra. The authors take considerable pains to rule out other confounding factors in the spectral analysis, which include wavelengths shifts in the O4 band, differences between experimental and theoretical spectra. The effects of different atmospheric structure on radiative transfer are also simulated. These experimental and analytical results are internally consistent within the uncertainties of the measurement. What may be valuable for future work on radiative transfer and theoretical studies on the water molecule absorption, is some discussion in the paper of whether it is possible to obtain more detailed experimental measurements of water absorption lines in the near-UV. In particular, would such an analysis be possible and more sensitive with a higher resolution system? Moreover, much of the initial impetus for measuring the water absorption spectrum was concerned with radiative transfer in the atmosphere. What implications does this paper bring to bear on that question? Possibly owing to the variety of topics explored, this is not an easy paper to read. Nevertheless, the standard of editing falls short of ACP standards and should be addressed. Some obvious errors are listed in the technical corrections, and I encourage the authors to review the text carefully again.

We would like to thank Referee #1 for the helpful comments. The comments helped us to improve the manuscript, showed up some missing details and improved the overall manuscript. We thank the reviewer for suggesting further points which could be studied in future studies. We agree that water vapour absorption in the blue and the UV needs further attention, in order to clarify the magnitude of currently known (and often used in various spectral retrievals, such as NO2 and IO) absorptions bands (see e.g. [Lampel et al., 2015]) as well as to quantify the magnitude of the absorptions in the UV. In view of the fact that the H2O lines are much narrower than our spectral resolution we agree with the suggestion of the reviewer that studies with higher spectral resolution could also be helpful. These measurements can ultimately also contribute further to the understanding of the water molecule which in turn will provide better models and thus better line lists. This step is then however clearly outside the scope of this manuscript.

Technical corrections:

1. Reported physical properties should have a space between the value and the units. This is not consistently adhered to in the manuscript.

We corrected these formatting issues (which appeared mostly for the wave number values).

2. Reference needed: p.4, l.22 after 'unaccounted tropospheric absorber'

We admit that this is a strong statement after various publications have used the O4 absorption in this spectral range. However, such a persistent residual structure can always point to 'unaccounted tropospheric absorber' if instrumental failures can be excluded, thus this is always a possibility. We split the sentence and the second part now reads: 'In any case, it could be possibly explained by an unaccounted tropospheric absorber'. We are not aware that this option was discussed in literature.

3. P.7, l.2-6: It is unclear whether the absorption cross section refers to total cross section, or to the differential cross section. The symbols used are those conventionally used for total absorption cross section. See e.g., Platt, Phys. Chem. Chem. Phys., 1999, 1, 5409-5415, for the usual description.

We refer here to total cross-section and changed the text to make this clear.

4. P23, 118: Do the authors have an explanation for the residual feature observed in one dataset?

We do not. However, as due to the almost constant humidity during M91 the dSCDs of  $O_4$  and  $H_2O$  and maybe other absorbers with similar concentration height profiles, a clear connection to water vapour absorption

cannot be established for this dataset. Therefore we concluded that this is not necessarily connected to water vapour absorption.

5. The quantities described in Table 5 are not sufficiently clear to this reviewer, and the columns should be more precisely defined than 'impact'. If, as I presume, what is meant is (e.g.) the difference between RMS (water absorption included) RMS (no water absorption), then this should be stated. Likewise for the other properties.

We added a short explanation to the caption of the figure.

6. The following parts of the document should be edited: Abstract: a. 'visible spectrum at a decreasing' . . . 'visible spectrum with decreasing'

**Corrected.**

- 7. 'until its dissociation limit' . . . 'up to its dissociation limit' **Corrected.**
- 8. Page 1: a. 15: 'vapour. it plays a key role for the'. . .' vapour. It plays a key role in the'

**Corrected.**

- 9. p1 16-17: 'Earth. . .absorption' . . .unclear. rewritten and shortened.
- 10. p1 19: 'also required assessing'. . .' also required for assessing' Corrected.
- 11. Page 4:19: 'and SO2, potentially even HCHO and BrO'. Unclear

Now we list all of the potentially affected trace gases, without weakening the statement for HCHO and BrO. This was written this way, as the water vapour absorption in the typical HCHO/BrO retrieval intervals could not be unambiguously identified in this publication.

12. Page 5: 20: 'Bremerhaven/. . ..employed'. Unclear

We reorded and shortened this sentence to clarify it.

- Page 7: a. 3: 'and narrow-band'. . .' and a narrow-band'
  Done.
- 14. b. 15: measurements is, that . . . measurements is that **Done.**
- 15. Page 8: a. 8: 'Longpath(LP)-DOAS' and 'here a a LASER-driven' Corrected.
- 16. 15: Unclear.

This is an advantage compared to MAX-DOAS measurements, a statement to this was added to the text. MAX-DOAS measurements use scattered light and therefore the light path length is initially not known and depends on a variety of factors, such as e.g. aerosol extinction profiles, viewing geometry and sun position.

- 17. Page 12:19: 'Due to need to' Corrected.
- Page 14: 2: 'selected such according'. Unclear Reordered and clarified.
- 19. Page 21: 27: 'water cross-section of O3 ,'. . .' absorption cross-section of O3 ,' Corrected.
- 20. Page 29: 29: 'strenghts' Corrected.

**2 Referee #2**

The paper discusses important new information about water vapour absorption in the UV spectral region and its effects on DOAS retrievals. With this it presents important results which are suitable for publishing in ACP. However, the presentation is at many points confusing and I suggest publication only after major revisions. Due to the lengths of this paper of 40 pages the content is hard to follow and this isnt helped by the fact that section titles dont always fit the content (e.g. section 4.9 is about the accuracy of the wavelengths axis and this should be spelled out in the title; section 1.3 lists science questions and not an outline). Another problem is that the authors clearly have lost track themselves, e.g.: there is no proper introduction about the differences of HITRAN 2009, 2012, HITEMP, or BT2 in the beginning of the manuscript, but there are bits and pieces of information later on in the text; the lower panels in Figure 1 are not referred to in the text at all; the text refers to a figure 4.2 which I believe is actually Figure 6; on the other hand, there is no reference in the text to the right panel in Figure 4; the spectral resolution of the instruments is stated 3 times in the manuscript, but some important information is only listed in captions, e.g. how the upper limit is calculated for Table 4: there are 2 different symbols used for absorber concentrations in the equations. More details/corrections below. However, I would like to encourage the authors to give this manuscript a thorough read themselves and to restructure some of its content, especially double-checking if the information provided in the figures/tables and their captions is actually used and sufficiently described in the manuscript.

We would like to thank Referee #2 for the positive remarks on the scientific value of our manuscript and the numerous helpful comments. They helped us to improve the structure of the manuscript to correct some inconsistencies and to clarify some points. Together with the revisions made in response to the comments of reviewer #1 these changes amount to a major revision of our manuscript.

Points addressed in the introduction:

1. 1. There is no proper introduction about the differences of HITRAN 2009, 2012, HITEMP, or BT2 in the beginning of the manuscript, but there are bits and pieces of information later on in the text

We added a several more sentences and some more references in the introduction of HITRAN, HITEMP and BT2. We moved the text part about the intensity line cutoff to the introduction. The details of how the line lists were created can be found in the given references. We changed 'HITRAN 2009' to 'HITRAN 2008 version 2009' due to a suggestion by Iouli Gordon.

2. 2. the lower panels in Figure 1 are not referred to in the text at all

We added these absorption cross-sections in order to illustrate the possibly affected spectral trace gas retrievals. We added a reference to this panel to the text when the potentially affected trace gas retrievals are listed..

3. 3. the text refers to a figure 4.2 which I believe is actually Figure 6;

This is correct. Many thanks for this hint! The reason was a wrong interpretation of the label in latex by the autoref command. This is fixed and refers now to section 4.2.

4. 4. on the other hand, there is no reference in the text to the right panel in Figure 4;

This is correct, as the text did not explicitly refer to the right panel. We added an explicit reference to this figure when the measurement error of the LPDOAS observations is discussed in the text.

5. 5. the spectral resolution of the instruments is stated 3 times in the manuscript, but some important information is only listed in captions, e.g. how the upper limit is calculated for Table 4

We moved the description from the caption to the text. We mentioned the spectral resolution in Table 4, as it differs for both instruments.

6. 6. there are 2 different symbols used for absorber concentrations in the equations.

**This is fixed.**

7. 7. section 4.9 is about the accuracy of the wavelengths axis and this should be spelled out in the title

Section 4.9 is about the accuracy of the wavelengths axis as well as about the shape of the absorption structures. If these are not represented well enough in the absorption line list, residual structures could have been observed. Both of these aspects are treated in this paragraph. We renamed this section to 'Estimation of the accuracy of the shape and wavelength calibration of the POKAZATEL  $H_2O$  cross-section'

Specific comments:

1. p.1, l.5, 11, 13: 363 nm or 362.3 nm. I understand that the authors refer to the peak of the absorption and the feature in general. But using two different numbers without further explanation in the abstract is confusing.

We changed this in the caption of Table 5, but left the 362.3nm value in the abstract unchanged as it describes the actual maximum of the absorption band at the given spectral resolution.

2. p.1, l.8: Add: For MAX-DOAS measurements, we observed. . .

**Added.**

3. p.1, l.8: It correlates. . . refers to something like 2 months of data. That should be made clear at this point.

A larger data set covering a longer time span could have been used, but it would not have changed the outcome of the study. We added 'The retrieved column densities from two months of measurement data and more than 2000 individual observations at different latitudes' to this sentence.

4. p.1, l.10: Add: . . .line intensities at 362.3 nm are underestimated by. . .

We added the wavelength of the absorption band in order to avoid confusion.

5. p1., l.12: spectral retrievals

Changed.

6. p.1, l.15: It

**Modified.**

7. Figure 1, top panel: The y-axis on the inset plot seems to have a different extent, especially at the lower end. This makes it appear as if the POKAZATEL has more lines in that inset than in the large plot.

Yes, the limits of the y-axis are different - at both sides. To avoid confusion, we added also labels to the inset plot.

8. p.3, l.6 and following: Why not refer to Figure 1 at this point already? The intro would read easier if you structured it like this: In Lampel et al. (2015b) you already suspected additional water vapour lines. But those could not be found in the line list available back then. Describe the available line list until then. Then you introduce the new POKAZATEL list and that this new information will be investigated with additional field measurements. Splitting the intro into subsections actually interrupts the flow of the argument.

We moved the part about the observations in Lampel et al. (2015b) to the top of the page and refer to Fig. 1.

9. p.3, l.16-17: Add that those were lab measurements.

Added.

10. p.3, l.20: structures in the spectral range

'systematic residual structures'  $\rightarrow$  'systematic structures in the fit residuals'

11. p3., l.24: individual line cut-off: what is that?

We added a reference to the HITEMP publication and added 'This cut-off removes weak absorption lines from the line list and was introduced for the HITRAN and HITEMP line lists to reduce the number of individual absorption lines for further processing as described e.g. in ... ' 12. p.3, l.28-29: No new paragraph needed here

**Removed.**

13. p.3, l.34: (in principle): Any explanation what this refers to or remove?

Removed. Referred to the fact that these absorptions are not yet observed and verified, but this is clear from the context.

14. p.4, l.15: I would remove potential here

Removed, added a reference to figure 1.

- 15. p.4, l.18,21: O4 was already used without being introduced as were the other species. We added the name for each trace gas, e.g. 'HCHO'  $\rightarrow$  'formaldehyde (HCHO)'
- 16. p.4, l.22: Please add reference for unaccounted tropospheric absorber.

We're not aware of any publication which reported this. However, residual structures in the spectral retrievals can always point to potentially neglected absorbers. We split the sentence, the second part is now 'In any case, it could be possibly explained by an unaccounted tropospheric absorber'. See also our response to reviewer #1 on the same topic.

17. p.4, 1.23-26: This statement at this point is difficult to understand for a person not very familiar with MAX-DOAS measurements and the corresponding radiative transfer. Either remove or give more explanation. See also below.

Removed here as it can be explained more nicely in the section on data evaluation.

- 18. p.5, l.2: formatting: brackets should be within the sentence. Changed.
- Section 2: Maybe add here that LP-DOAS is active and MAX-DOAS a passive technique We added passive and active to this section.
- 20. p.5, l.12: remove space after 15. Removed.
- 21. p.5, l.15: Full stop after  $0.45\mathrm{nm}.$

**Added.**

22. p.5, l.16: The latitudinal extent has nothing to do with variations of the water vapour mixing ration. Also, Figure 2 is misleading since the satellite in the background is from a different time than the M91 cruise.

We agree that latitude and water vapour mixing ratio are per se independent, but the maximum absolute water vapour content of air depends strongly on temperature. Therefore strong latitudinal gradients in water vapour mixing ratio can be seen, e.g. during ANT XXVIII/1-2. We added that the cruise M91 was additionally also a short one, limiting the variation of observed water vapour mixing ratios.

**23. p.5, l.17-19: S.a.**

Here we added an explanation about why  $O_4$  and water vapour are correlated for measurement conditions with small variations in absolute humidity: 'The  $O_4$  dSCD is as a first order approximation proportional to the effective light path length, the H2O dSCD is proportional to the light path length as well, but also to the absolute humidity along the light path according to Eq. 2. '

24. p.5, l.20: therefore: s.a. water vapour changes because it changes and not because the measurements are at a different latitude.

We removed 'therefore'.

25. p.5, l.22-24: Maybe make that 2-3 sentences. Your point that O4 and H2O absorb in similar region in the UV as well as in the visible doesn't fully come across. You could also refer to Figure 1 here.

We split the sentences and referred again to figure 1.

26. p.5, l.3: lambda is not introduced

**We added a short introduction for the wavelength $\lambda$ .**

27. Figure 2: Its not clear how you get from a slant column ratio to a vertical column of one of the species. Also these results are not discussed in the manuscript.

Figure 2 is intended to be an overview about the measurement locations. We added a short paragraph to the text: 'In Fig. 2 the ratios of  $H_2O$  and  $O_4$  dSCDs at 3 telescope elevation were converted to  $H_2O$  VCDs assuming a lightpath at ground level under normal conditions and a water vapour scale height of 2 km and using the correction factor of 2.6. Qualitatively the latitudinal variation of the ANT XXVIII/1-2 and GOME-2 data agree. For a quantitative comparison further radiative transfer modelling to obtain tropospheric water vapour profiles from the ship-based data would be needed.' Further comparisons of VCDs are outside the scope of this manuscript. The caption of Fig. 2 already contains the time of the averaged GOME-2A VCDs.

28. p.7, l.1: I0(lambda) is introduced a second time

**Removed this.**

29. p.7, l.2: OD is not introduced yet

Added.

30. p.7, l.3: why only partly?

Partly was indeed the wrong word here, we reformulated to 'The measured OD of the broad-band extinction and scattering by molecules and particles is represented by a polynomial'

31. Eq (1), (2): please use same symbol for concentrationDone.

32. Eq (1): add a bracket to indicate the summation; the polynomial p(lambda) is a different one than the one introduced in the line 3 above for the measurements and the cross section here should be a differential cross section

Formally, we think no brackets are needed here. The polynomial is the same as mentioned in line 3. We added here, following a suggestion from the first reviewer, that we refer here to the total absorption cross-section  $\sigma$ .

33. p.7, l.9: See above; maybe add somewhere before already that MAX-DOAS measures scattered sunlight and LP-DOAS is an active technique.

We added this at the introduction of the instruments.

- 34. p.7, l.12: spectral width **Done.**
- 35. p.7, l.14-15: The sentence about the residual is confusing at this point. Maybe remove?

We changed 'residual' to 'measurement error of slant column density'. This way we can point out the advantage of the MAX-DOAS measurements without using the word 'residual' in this context.

36. p.8, l.4-6: The total light path is from the institute to the train station and back to the institute?

Yes. We added that the light also travels back from the retro reflector to the telescope.

37. p.8, l.6: The spectral resolution is redundant information here. Was mentioned before. Section 3.1: I suggest swapping the first 2 paragraphs.

Swapped the two paragraphs.

38. p.8, l.11: Not the measurement sequence but the correction with the background spectra ensures the independence.

We split the sentence and added that the measurement spectra are corrected explicitly with the background spectrum.

39. p.8, l.17: high-pass filtered literature cross-sections: aha! That should be mentioned before.

As this is only the case for the LP-DOAS measurements, this is mentioned here and not at the general DOAS description. The MAX-DOAS data is not filtered and only the polynomial is applied.

40. p.8, l.27: s.a., symbol for concentration.

s.a., already changed.

41. p.8, l.31: A Fraunhofer spectrum always refers to the extra-terrestrial spectrum of the sun (or another star).

As it seems to be usual to name the reference spectrum for the MAX-DOAS evaluation Fraunhofer Reference (as also in Platt and Stutz 2006), we added here once 'A so-called Fraunhofer reference spectrum (we follow the customary nomenclature to call such a spectrum Fraunhofer spectrum although it also contains spectral features from Earth's atmosphere)' and continue to use this name later on in the manuscript. As ground-based MAX-DOAS instruments have no chance to measure an extra-terrestrial spectrum of a star, typically a spectrum with only small absorptions is used as the so-called 'Fraunhofer reference spectrum'. This is often a zenith sky spectrum, as that is typically a spectrum with the smallest amount of atmospheric absorption which can be recorded with the given instrument. As MAX-DOAS instruments are typically not radiometrically calibrated (see e.g. [Wagner et al., 2015] and [Lübcke et al., 2016]), and thus the instrument response function is not perfectly known. Since the solar atlases often still contain telluric absorption lines (compare e.g. the data from [Kurucz et al., 1984] and [Chance and Kurucz, 2010]), it is often better (in terms of minimising the fit residuals) to use a reference spectrum recorded by the same instrument for the spectral retrieval. This is especially important for the detection of weak absorbers.

42. Table 2: What is this Add. Polynomial degree?

The additional polynomial is used in spectral retrievals of MAX-DOAS data to compensate instrumental stray light and usually neglected effects, as e.g. vibrational Raman scattering (VRS, Lampel et al 2015). We added a sentence to the description of the spectral retrieval of MAX-DOAS data: 'An additional intensity offset polynomial was used in the spectral evaluation to compensate for instrumental stray light, as described e.g. in [Peters et al., 2014].' An overview of different implementations can be found in [Peters et al., 2016].

43. p.10, l.1: Fraunhofer: s.a. and another time below as well.

s.a.

- 44. p.10, l.2: full stop after bracket **Done.**
- 45. p.10, l.3: Remove the last the of the line **Done.**
- p.10, l.6: ANT XXVIII/1-2 or ANT XXVIII? Please unify in manuscript.
  Done.
- 47. p.10, l.16: spectral resolution is redundant information

**Removed.**

48. p.10, l.20: Why Figure 5 before Figures 3 & 4?

As the LP-DOAS measurements yield direct concentration values along the lightpath with the need to consider radiative transport, it was decided to start with the LP-DOAS measurements instead of the MAX-DOAS observations. Therefore the nicer fits (figure 5) are found after the LP-DOAS data

for the MAX-DOAS data. We added to the introduction of the results, that the LP-DOAS data 'have the advantage of a well-defined light path length' and are therefore presented firstly.

49. p.10, l.21: 40 telescope angle.

Changed.

- 50. p.10, l.21: Spectra recorded at. . .: why not remove the sentence in l.1-2, p.10 then? Good idea, done.
- 51. p.10, l.25: Are those the dSCD measurement errors? Please clarify. Also state that this disregards possible systematic errors.

Added: 'This estimate potentially disregards possible systematic errors, but these are estimated to be small compared to the water vapour absorption  $(< 2 \times 10^{-4})$  as the residuals are dominated by random shot noise (see Fig.5)'.

52. p.10, l.26: This section is about the DOAS spectral fitting. So a reference to a linear fit is confusing here. Please add more explanation or move/remove this sentence.

We added '... the residual of the linear fit of  $H_2O/O_4$  ratios at 363 and 477nm shown in ...'.

53. p.11, l.25: OD; s.a.

Explanation added above.

54. p.11, l.28-30: Please elaborate or state reference.

First of all we added a small introduction to this paragraph (with references) in order to introduce the Ring effect itself: 'The Ring spectrum itself compensates the measured apparent optical density due to inelastic scattering of sunlight at air molecules [Shefov, 1959, Grainger and Ring, 1962], which leads to an effective filling-in of Fraunhofer lines in the measured spectrum of scattered sunlight e.g. [Wagner et al., 2009] and references therein.'

Further elaboration such as RTM for the effective temperature of the Ring effect would be out of the scope of this work. We added an estimate of the total magnitude of this effect:'For a Ring signal of  $2.5 \times 10^{25}$  molec cm-2 (which is typical for MAX-DOAS observations), the temperature dependence of the Ring effect results in an OD of  $5 \times 10^{-4}$  for a temperature difference of 30 K. In our analyses warmer effective Ring temperatures were found at low telescope elevation angles.' We did also run the evaluations again without correction of the Ring temperature effect and found no significant changes of the overall result regarding the size of the water vapour absorption around 363nm. It led however to elevation angle separated systematic residual structures as found by PCA analysis (similar to [Lübcke et al., 2016]) of the resulting fit residual spectra and was therefore included in the final analysis.

55. p.12, l.3: when co-adding spectra from more than Modified.

56. p.12, l.12: Remove paragraph break.

Done.

57. p.12, l.16: Is the stated time period different from the one in Table 1?

Only a subset of the measurements was used. During other days, e.g. the short-cut measurements failed. Therefore we reduced the dataset to those measurements where optimal conditions were found. We added 'when optimal instrumental performance could be guaranteed'

58. p.12, l.20: Are you really losing 15 min for each hour? Please clarify.

No, background measurements are performed for lamp reference measurements as well as for atmospheric measurements, which means that four spectra are recorded during each sequence. We added to the paragraph about the background correction, that these are four spectra in total. The time to change the wavelength range is negligible in this configuration and these exposure times.

59. p.12, l.25: The data in Figure 3 does not support this statement. Also, this is the only time Figure 3 is mentioned in the text of the manuscript (besides in the caption for Figure 2, but Figure 2 is barely mentioned either. Those would be candidates for removing in order to shorten the manuscript.).

Adding more spectra was tested, but did not yield satisfying results or improved the results. Longer times for co-adding spectra increases also the effect of potential instrumental changes. We decided to keep the figures to show the results of the LP-DOAS measurements.

60. p.12, l.29: 20% is not low humidity?

This is not necessarily low absolute humidity during summer, compared to the overall data set. These values were observed around noon with high outside temperatures.

61. p.13, l.2: Maybe remove the uncertainty estimate at this point since it has just been stated in the line above and the actual interesting number is 0.7 and not 0.05.

done

62. Figure 4, caption: 2.31? The text states 2.4. The right panel of the figure is not mentioned in the text at all.

2.31 is the factor when allowing an y-axis intercept in the fit, 2.39 when no y-axis intercept is allowed for. This is already stated in the text. The right panel is now mentioned explicitly.

63. p.14, l.1-3: Please rephrase.

Sentence split and modified: 'These fitting intervals were selected in a way, that the wavelength of the main absorptions of  $O_4$  and  $H_2O$  are at similar wavelengths. This needs to be done in order to have approximately the same radiative transfer properties for both absorbers'

64. p.14, l.10: The figure states R2 = 0.74 for both cases. Please clarify!

Thanks for pointing this out. We checked the script and updated the plot. The wrong variable was written to the plot, but the correct data is found in the output for the table of results using different O4 XS.

65. Figure 6, caption: There is only 1 error bar and that is attached to the linear fit. Does it refer to the error bars of the measurements though? Please clarify!

The errorbar represents the mean measurement error for all considered measurements, it is now further clarified in the caption.

66. For the green box in the top right panel, how are the measurement uncertainties combined? The figures say O4 at 476nm and not 477 as in the caption.

We replotted the figure using consistent wavelengths. The green box represents twice the mean DOAS fit error for the measurements as stated in the caption of the figure.

67. p.17, l.5-7: I dont understand this paragraph. Worse in comparison to what? Did you state the number of the combined correlation somewhere? 0.91 is a pretty good correlation.

Worse in comparison to the correlation of their respective ratios. Added 'compared to the correlation of their respective ratios'

68. p.17, l.8-11: This information should already be stated on p.14,l.9-10. Also maybe mention somewhere that this is the reason for the different numbers for n in Table 3.

Moved. We added 'These conditions lead to different numbers of valid observations in Table 3 for different spectral retrieval settings'.

- 69. p.17, l.14: see also Table 3 changed
- 70. p.17, l.16: latitudinal s.a.

removed here.

- 71. p.18, l.16: includes more measurements: see comment above See comment above.
- 72. p.18, l.31: could not be identified for either of the two line lists or cruises? Please clarify! This was tested for M91. We added this. ANT XXVIII/1-2 data was also analysed, but not included in the manuscript in this case as it did not yield further information and larger detection limits.
- 73. Figure 7, caption: add space after DMS

Done.

74. p.19, l.12: an RMS Changed.

- 75. p.19, l.15: than in either the BT2 or the HITEMP
  - changed to 'to be better predicted in the *POKAZATEL* line list than in the BT2 and the HITEMP line list'
- 76. p.19, l.15-18: This information should have been in the intro.

Shortened the sentence here and added to the introduction.

- 77. Figure 8: gridlines would be helpful in this figure.We added gridlines to this figure.
- 78. p.22, l.18: 3 elevation angle

Added

79. p.22, l.16: polynomials with degrees 0-2 were applied in order to test. . .

Changed. We added also '... to estimate the dependence of the inferred upper limits on the degree of the DOAS polynomial. The polynomial could compensate for water vapour absorption if it would be a rather broad absorption in this spectral region as suggested by [Du et al., 2013]'

80. Table 4: Last sentence of the caption seems to be quite important, however, is not explained in the text.

This sentence was moved from the caption to the text.

81. Section 4.9, title: Please add that you investigate the accuracy of the wavelength calibration here

Modified to 'Estimation of the accuracy of the shape and wavelength calibration of the POKAZATEL  $H_2O$  cross-section' (see above)

82. p.23, l.7: why is there an R introduced for the residual spectra? Its not used anywhere else.

**Removed.**

83. p.24, l.5: Within 0.1 cm-1 in comparison to what?

Compared to laboratory measurements. We added 'from measured data from [Maksyutenko et al., 2007]'

84. p.25, l.1: Section title is misleading. This section only refers to the visible range.Added 'in the blue wavelength range'. This is important, as these were used

as a 'reference' to compare to the UV data.

85. p.25, l.5: . . . magnitude of the water vapour. . . blue wavelength range. . .

We restructured this sentence to 'The uncertainty of the absolute magnitude of the water vapour cross-section (HITEMP) in the blue wavelength from 452-499 nm is less than 15%: ...'.

86. p.25, l.5-8: Where does this information come from?

We added 'when fitting the absorption bands separately analogously to [Lampel et al., 2015]' to the end of the sentence.

- 87. P.25, l.14: formatting issue H2O-dSCD Fixed.
- 88. p.25, l.21-23: Please split sentenceSplit sentence and removed one of the 'which'.
- p.25, l.27-29: Please split sentence
  Done.
- 90. p.25, l.30-31: No direct correlation. . . I dont understand this. Please elaborate.

We meant 'No correlation of the water vapour dSCDs at 363 nm with the square term of the  $\rm O_4$  absorption was found for the ANT XXVIII/1-2 dataset.' This is now corrected

- 91. p.26, l.1-3: Maybe use the term water vapour contamination here.Good idea, changed.
- 92. p.27, l.2: formatting issues for references **Fixed.**
- 93. p.27, l.4: formatting issues for reference **Fixed.**
- 94. p.27, l.20: section 1.2 does not mention the correction factor **Removed.**
- p.27, l.30: formatting issues for reference and brackets
  Fixed.
- 96. p.28, 1.5-8: Why didnt you perform this analysis separately then for cases with and without HONO?

The HONO absorptions are close to the detection limit, which make a simple filtering difficult or impossible. We tried to filter based on the NO2 dSCDs, but as large NO2 dSCDs introduce again residual structures, these would have to be filtered out as well. Finally such a pre-filtered data set would have looked a bit arbitrarily filtered. Therefore we used the complete dataset using only the RMS as quality indicator. The positive value of the mean HONO dSCD is however in agreement with zero, it is within the standard deviation of the observed values  $(1.0 \pm 2.3) \times 10^{14}$  molec cm-2. We added to the manuscript '..., but it is in agreement with zero within the standard deviation of the observed values. Filtering the results based on HONO dSCDs could have introduced a negative bias, as the observed HONO values are generally close to the respective detection limits. We therefore used the complete MAX-DOAS data set.'

Furthermore we added 'Only fit results with an initial RMS of the fit residual of less than  $4 \times 10^{-4}$  were considered' to the introduction of this subsection.

97. p.29, l.1: alternative to what?

Alternative to the standard evaluations, which are described in ([Bobrowski et al., 2010, Hörmann et al., 2013]). However, we removed 'alternative'.

98. p.29, l.7: remove itself

Removed.

99. p.29, l.16: rephrase sentence

'in the blue spectral' $\rightarrow$ ' in the blue spectral range'

100. p.29, l.22: Maybe join these two sections?

We prefer to keep the sections separated, as one involves the detection of water vapour absorption in the UV, while the other is an addition to [Lampel et al., 2015].

101. Figure 11, caption: different bands listed in Table6; use unity instead of 1.

Changed.

102. Table 6, caption: relative integrated absorption values: relative to what? Please elaborate;

'relative integrated absorption values'  $\rightarrow$  'integrated absorption values relative to W3'

103. The second to last, not last row shows the scaled HITEMP data.

The last row was removed in the final version of the manuscript, but not from the caption of the table. We removed it also there.

- 104. p.31, l.10: Who are they? changed to 'these'.
- 105. Table 7, caption: formatting issues with reference

Changed.

106. Appendix: Why are table 7 and 8 in an appendix? Then the text discussing those should also be moved to the appendix.

We moved both tables to the part of the text which discusses these results.

107. Table 8: Im not sure what is done here. What are the relative DOAS fit errors? There is no Window 5 in here.

These two tables are added in analogy to [Lampel et al., 2015], as extension of those. It uses the same MAX-DOAS datasets, but POKAZATEL was not available back then.

More general comments:

108. More general comments: \* The abstract doesn't list anything about the O4 studies or the failed identification of other water vapour lines in the UV

We added that different  $\rm O_4$  absorption cross-sections were tested: 'The results were independent of the used literature absorption cross-section of  $\rm O_4$ , which overlays this water vapour absorption band'

109. More general comments: \* References to dissertations cannot be accessed if no link is provided.

We added a URL in both cases.

**References**

- [Bobrowski et al., 2010] Bobrowski, N., Kern, C., Platt, U., Hörmann, C., and Wagner, T. (2010). Novel so2 spectral evaluation scheme using the 360-390 nm wavelength range. Atmospheric Measurement Techniques, 3(4):879–891.
- [Chance and Kurucz, 2010] Chance, K. and Kurucz, R. (2010). An improved high-resolution solar reference spectrum for earth's atmosphere measurements in the ultraviolet, visible, and near infrared. Journal of Quantitative Spectroscopy and Radiative Transfer, 111(9):1289 – 1295. Special Issue Dedicated to Laurence S. Rothman on the Occasion of his 70th Birthday.
- [Du et al., 2013] Du, J., Huang, L., Min, Q., and Zhu, L. (2013). The influence of water vapor absorption in the 290-350nm region on solar radiance: Laboratory studies and model simulation. *Geophysical Research Letters*, 40(17):4788–4792.
- [Grainger and Ring, 1962] Grainger, J. and Ring, J. (1962). Anomalous fraunhofer line profiles. Nature, 193:762.
- [Hörmann et al., 2013] Hörmann, C., Sihler, H., Bobrowski, N., Beirle, S., Penning de Vries, M., Platt, U., and Wagner, T. (2013). Systematic investigation of bromine monoxide in volcanic plumes from space by using the gome-2 instrument. *Atmospheric Chemistry and Physics*, 13(9):4749–4781.
- [Kurucz et al., 1984] Kurucz, R. L., Furenlid, I., Brault, J., and Testerman, L. (1984). Solar Flux Atlas from 296 to 1300 nm. National Solar Observatry, Sunspot, New Mexico, U.S.A.
- [Lampel et al., 2015] Lampel, J., Pöhler, D., Tschritter, J., Frieß, U., and Platt, U. (2015). On the relative absorption strengths of water vapour in the blue wavelength range. Atmospheric Measurement Techniques, 8(10):4329–4346.
- [Lübcke et al., 2016] Lübcke, P., Lampel, J., Arellano, S., Bobrowski, N., Dinger, F., Galle, B., Garzón, G., Hidalgo, S., Chacón Ortiz, Z., Vogel, L., Warnach, S., and Platt, U. (2016). Retrieval of absolute so2 column amounts from scattered-light spectra – implications for the evaluation of data from automated doas networks. *Atmospheric Measurement Techniques Discussions*, 2016:1–33.
- [Maksyutenko et al., 2007] Maksyutenko, P., Muenter, J. S., Zobov, N. F., Shirin, S. V., Polyansky, O. L., Rizzo, T. R., and Boyarkin, O. V. (2007). Approaching the full set of energy levels of water. *Journal of Chemical Physics*, 126(24):241101–241101.
- [Peters et al., 2016] Peters, E., Pinardi, G., Bosch, T., Richter, A., Wittrock, F., Seyler, A., Burrows, J. P., Roozendael, M. V., Hendrick, F., Drosoglou, T., Bais, A., Kanaya, Y., Zhao, X., Strong, K., Lampel, J., Frieß, U., Volkamer, R., Koenig, T., Ortega, I., Dix, B. K., Piters, A., Puentedura, O., Navarro, M., Gomez, L., Gonzalez, M. Y., Remmers, J., Wang0, Y., Wagner, T., Wang, S., Saiz, A., Nieto, D. G., Cuevas, C. A., Querel, R., Johnston, P., Postylyakov, O., Borovski, A., Bruchkouski, I., Li, C., Hong, Q., Rivera, C., Grutter, M., Stremme, W., Khokhar, F., and Khayyam, J. (2016). Investigating differences in doas retrieval codes using mad-cat campaign data. Atmospheric Measurement Techniques Discussions, prepared for publication.

- [Peters et al., 2014] Peters, E., Wittrock, F., Richter, A., Alvarado, L. M. A., Rozanov, V. V., and Burrows, J. P. (2014). Liquid water absorption and scattering effects in doas retrievals over oceans. *Atmospheric Measurement Techniques*, 7(12):4203–4221.
- [Shefov, 1959] Shefov, N. N. (1959). Intensivnosti nokotorykh emissiy sumerochnogo i nochnogo neba (intensities of some emissions of the twilight and night sky). Spectral, electrophotometrical and radar researches of aurorae and airglow, IGY program, section IV, 1:25.
- [Wagner et al., 2015] Wagner, T., Beirle, S., Dörner, S., Penning de Vries, M., Remmers, J., Rozanov, A., and Shaiganfar, R. (2015). A new method for the absolute radiance calibration for uv-vis measurements of scattered sunlight. *Atmospheric Measurement Techniques*, 8(10):4265–4280.
- [Wagner et al., 2009] Wagner, T., Deutschmann, T., and Platt, U. (2009). Determination of aerosol properties from MAX-DOAS observations of the Ring effect. *Atmospheric Measurement Techniques*, 2(2):495–512.

---

## Author Response (AR2)

**We would like to thank the reviewer for his helpful comments on the revised manuscript. We changed the manuscript accordingly and hope that all points are now addressed in a satisfying way.**

The following numbers refer to the previous review of reviewer #2

Points addressed in the introduction:

3. There are still 2 instances where the authors refer to a figure 4.2 instead of the section 4.2

**Again another autoref LaTeX issue. Thanks for pointing this out, we changed this.**

5. I'm not sure if I'm old-fashioned here, but there is no sentence in the manuscript stating that the spectral resolution and other instrumental information is given in Table 1.

**We added a reference to Table 1 after listing the campaigns: 'An overview of the instruments used is given in Table 1.'**

Specific comments:

54. DOASIS wasn't introduced as a DOAS analysis package. There is only a reference given in one of the tables as Krauss (2006). However, the link provided in the literature section for this reference doesn't work. It seems to be an internal webpage.

**We now introduce DOASIS as the DOAS analysis package which was used for LPDOAS and MAX-DOAS data. 'The spectral analysis was done using the DOASIS software package \citep{Kraus2006}.'**

58. The text in the reply is clearer than the text that was added to the manuscript.

**We reformulated the paragraph describing the LP-DOAS measurements.**

59. I still don't think that Figure 3 supports the statement. I only see a structured residual. But I can't assess from this if those structures are systematic from 1 example.

**The residual is clearly not dominated by shot noise (which would be the ideal case for summing more spectra), this can be seen from Figure 3. Residual spectra which are closer to shot noise can be seen e.g. for the UV MAX-DOAS fits shown in Figure 5.**

62. Seems to be unnecessarily confusing to state 2.31 in the caption if the important number is 2.4.

**We removed this number from the caption of the plot to avoid confusion. It now reads '[...] corrected by the scaling factor determined from the correlation [...]'.**

65-66. I think the authors make it appear more trivial than this is. Did you add the individual uncertainties in quadrature? Did you double the combined uncertainties or the individual ones for the overall 2sigma uncertainty. I think this needs more explanation.

**You are correct that all errors need to be added in quadrature. In this case however the statistical error of the linear fit (<0.01 / 2.63) is significantly smaller than the systematic error for the water vapour dSCD in the UV around 363nm, which is estimated to be around 8% in Table 3. Therefore the DOAS fit error dominates clearly. We added 'The contribution of the statistical error of the linear fit is negligible' to the caption of the figure.**

72. This sentence still doesn't make sense. Shouldn't it be: 'could not be identified for either of the other two line lists'?

**We changed this sentence to 'These could however not be identified for either of the two \POKAZATEL nor the BT2 line lists during the M91 cruise' in order to clarify this explicitly.**

109. Not for Eger (2014) and Bussemer (1993) and the link to Krauss (2006) doesn't work

**These three publications are not available online, but on request at the University. The work by Bussemer (see also http://katalog.ub.uni-heidelberg.de/titel/59803137 ) is mostly found also in the DOAS book, which is why we already state in the revised manuscript 'from Bussemer (1993), parts of which can also be found in Platt and Stutz (2008)'.**
**The work by Kraus is also published by Shaker in Aachen, Germany in 2006 and can be obtained there. We added ISBN and publisher to the bibtex entry.**

66: For figure 6, the authors only changed the labels on 2 of the 3 plots and the third one still has wrong labels.

**The ratio of dSCDs and the $H_2O$ dSCDs are given at the center wavelength of the water vapour absorption, the O4 dSCDs are given at the at the center of the O4 absorption in the fit interval. As discussed in 3.3.1. the fit intervals were chosen in such a way that the center wavelengths of both absorptions are as close together as possible in the VIS as well as in the UV. The difference of the center wavelengths of both absorbers is 2nm in the UV and 3nm in the visible.**

[revised manuscript text omitted]